# Viability leads to the emergence of gait transitions in learning agile quadrupedal locomotion on challenging terrains

Milad Shafiee [1] ✉, Guillaume Bellegarda [1] & Auke Ijspeert [1]

Quadruped animals are capable of seamless transitions between different gaits. While energy efficiency appears to be one of the reasons for changing gaits, other determinant factors likely play a role too, including terrain properties. In this article, we propose that viability, i.e., the avoidance of falls, represents an important criterion for gait transitions. We investigate the emergence of gait transitions through the interaction between supraspinal drive (brain), the central pattern generator in the spinal cord, the body, and exteroceptive sensing by leveraging deep reinforcement learning and robotics tools. Consistent with quadruped animal data, we show that the walk-trot gait transition for quadruped robots on flat terrain improves both viability and energy efficiency. Furthermore, we investigate the effects of discrete terrain (i.e., crossing successive gaps) on imposing gait transitions, and find the emergence of trot-pronk transitions to avoid non-viable states. Viability is the only improved factor after gait transitions on both flat and discrete gap terrains, suggesting that viability could be a primary and universal objective of gait transitions, while other criteria are secondary objectives and/or a consequence of viability. Moreover, our experiments demonstrate state-of-the-art quadruped robot agility in challenging scenarios.

Quadruped animals learn impressive abilities to traverse challenging terrain, reach remote parts of the planet, and perform agile locomotion in pursuit of prey. They learn to avoid falling in all of these locomotion scenarios, which in robotics terms means staying viable. Such fascinating locomotion skills emerge from inter-limb coordination governed by the interaction between the brain, the spinal cord, and the musculoskeletal system[1]. The modulation of this inter-limb coordination can produce different gait patterns, and the transition between these patterns is a fundamental feature of locomotion. Quadruped animals smoothly transition between different locomotion gaits such as walking, trotting, galloping, and bounding/pronking depending on their speed. However, despite the growing number of studies on gait transitions in both robotics and biology, there is still no clear consensus regarding the underlying mechanisms as well as the criteria that determine why gait transitions take place in different conditions.

Different quantities have been proposed as important in gait transitions: energy expenditure, peak forces, and periodicity. A study conducted by Hoyt and Taylor[2] suggested that, when changing locomotion speed, horses switch gaits to reduce energy expenditure. While energy efficiency is now the most widely accepted objective for gait transitions, other studies have found that this hypothesis may not always be valid for humans[3,4]. As an alternative to the energy-efficiency hypothesis, Farley and Taylor[5] have suggested that the trot-gallop gait transition in horses is triggered when peak musculoskeletal forces reach a critical level. In particular, transitioning between gaits makes it possible to reduce peak musculoskeletal forces (and consequently the risk of injury). However, at the transition speed, galloping requires more energy than trotting, meaning that this gait transition increases the Cost of Transport (CoT). Therefore, the second suggested criterion for gait transitions is reducing peak musculoskeletal forces. Granatosky et al.[6] observe that, across nine species, gait periodicity

[1]Biorobotics Laboratory, École Polytechnique Fédérale de Lausanne (EPFL), 1015 Lausanne, Switzerland. ✉e-mail: milad.shafiee@epfl.ch

(computed by measuring inter-stride variability) predicts the occurrence of gait transitions better than energy efficiency. Their findings suggest that gait transitions are performed to maximize periodicity (i.e., minimize inter-stride variability) and avoid unstable gaits. They link periodicity to stability, and hypothesize that periodicity (and hence stability) is another important gait transition criteria on flat terrain. High periodicity can furthermore be useful for anticipatory behavior on flat terrain by simplifying the prediction of future states (e.g., future foot placements). However, periodicity is not necessarily a good feature per se, in particular when the terrain is irregular.

Animal experiments have played a crucial role in exploring the criteria for gait transitions in the aforementioned research. In particular, experiments with spinal transections and decerebrations in cats have been instrumental in establishing the functional roles of the neural circuits in the spinal cord during locomotion[7,8]. For example, physiological studies have found that electrical stimulation or increasing the speed of a motorized treadmill can cause a decerebrated cat to spontaneously change gaits[9–11]. However, such animal experiments clearly raise ethical issues, making extensive data collection impossible. As an alternative, we can use robots as tools to investigate scientific hypotheses about animal motor-control, iteratively taking inspiration from animals to improve the robots' performance[12–16]. Robots allow us to test computational models of the spinal cord in closed-loop to measure internal states during agile locomotion on challenging terrains, which is not possible with animals.

In robotics, abstract models of the spinal cord composed of central pattern generators (CPGs), i.e., systems of coupled oscillators, and reflexes are commonly used for locomotion pattern generation[17–20] as well as for the investigation of biological hypotheses[21,22]. CPGs provide an intuitive formulation for specifying different gaits[23], and spontaneous gait transitions can arise by increasing descending drive signals and incorporating contact force feedback[24,25] or vestibular feedback[26]. Within a dynamical systems context, it has been suggested that gait transitions serve to avoid unstable states[27–29]. Gaits can be viewed as a result of complex systems that self-organize around their natural dynamics, and gait transitions occur when the stability of the system dynamics decreases so much that switching to a new gait increases stability. With this dynamical systems analysis, the CoT is described as a surrogate for the stability of the underlying dynamics, but it is not considered as a primary determinant per se, since gaits with lower stability necessitate active control for stabilization, which consequently results in higher energy consumption. Recent studies have shown the possibility of acquiring different gaits through deep reinforcement learning (DRL)[30–36]. While gait transitions did not autonomously emerge within these learning frameworks, the walk-trot transitions were found possible through a combination of model predictive control (MPC) and DRL by minimizing energy consumption[37].

In summary, animal and robotics investigations suggest so far that energy efficiency, stability, and avoiding peak forces (injury) to the musculoskeletal system are plausible explanations for animals to transition between different gaits. In this article, we explore the potential role of another criterion: the concept of viability, which formalizes the notion of avoiding a fall during legged locomotion. Viable states are all of the states starting from which a system can avoid falling through proper action selection[38]. Viability is a useful general objective for locomotion control, and it is influenced by multiple factors such as periodicity, gait stability (in a Lyapunov sense), and negative events such as falls into a gap or collisions with obstacles. While viability is related to periodicity and gait stability, it is a broader concept. In this article, we use the general concept of viability to describe gait stability and periodicity during locomotion on flat terrain, as well as fall avoidance during gap-crossing scenarios.

We use a hierarchical biology-inspired framework leveraging robotics and DRL tools[39] to investigate the following research questions surrounding the emergence of quadruped gait transitions:

1. What are the plausible determinants for quadruped animal gait transitions from the following options: minimization of Cost of Transport (CoT), minimization of peak forces, and/or viability?
2. Given that animals such as horses use certain gaits at specified velocity ranges, what is the explanation for the shallower CoT-Velocity plot of "normal" walk/trot gaits, compared with extended walk/trot gaits (i.e., when using a gait outside of nominally trained operating regions)?
3. Does the environment (i.e., a terrain with gaps) impose certain gaits, and how does anticipatory sensing of upcoming terrain affect gait transitions? What is the relationship between gait transitions imposed by the terrain, and the aforementioned criteria of CoT, peak contact forces, and viability?
4. What kind of exteroceptive sensory information is most effective (and sufficient) for triggering gait transitions when a legged system must cross a terrain with gaps?

To answer these questions, we investigate the interaction between supraspinal drive and a CPG to produce anticipatory locomotion for a quadruped robot[39]. Using DRL, we train a neural network policy to replicate the supraspinal drive behavior. This policy can either modulate the CPG dynamics, or directly change actuation signals to bypass the CPG dynamics (Fig. 1).

## Results

In this section, we investigate and present locomotion results in two environmental scenarios: one for flat terrain and another for discrete gap terrain. In the discrete gap terrain, we investigate the objective for gait transitions by considering three main elements in the reward function: viability, CoT, and peak contact forces. When training the locomotion policies, we consider the effects of four different values (high, medium, low, and zero) for the reward weights. Our analysis of these reward function term weights reveals that the combination of high, low, and low weights for viability, CoT, and peak forces, respectively, yields the best performance (highest gap crossing success rate).

On flat terrain, we train distinct policies for our robot to learn locomotion at various velocities, focusing on specific gaits such as walk and trot. To achieve different velocities, we incorporate a velocity tracking term into the reward function. On flat terrain, each gait is achieved by utilizing explicit oscillator phase coupling matrices in the abstract CPG equations, specifically defined for walk and trot (see Methods and Supplementary Material). In this scenario, along with the velocity tracking term, our reward function gives a high weight to viability and a low weight to energy consumption, similar to the best gap crossing scenario result. We do not include the peak force component in the reward function for blind locomotion, as our investigation on flat terrain is centered around walk-trot gaits. The reduction of musculoskeletal forces has been studied in the context of the trot-gallop gait transition in horses, particularly at high velocities[5]. However, for the walk and trot gaits at normal velocities, critical peak contact forces are limited and hence are not included in the reward function for flat terrain.

### Comparison of robot and animal data for flat terrain locomotion

We examine the consistency of our hierarchical biology-inspired learning architecture with animal locomotion experiments performed by Granatosky et al. which characterized how gaits, energy efficiency, and periodicity depend on speed[6]. Granatosky et al. estimated the CoT based on oxygen consumption: the lower the CoT, the higher the energy efficiency. They computed periodicity from the Coefficient of Variation (CV) of the stride duration: the lower the CV, the higher the periodicity. Figure 2A presents results for the first experimental category: walking and trotting for the robot with data from four quadruped animals: domestic dog (*Canis lupus familiaris*), Australian water rat

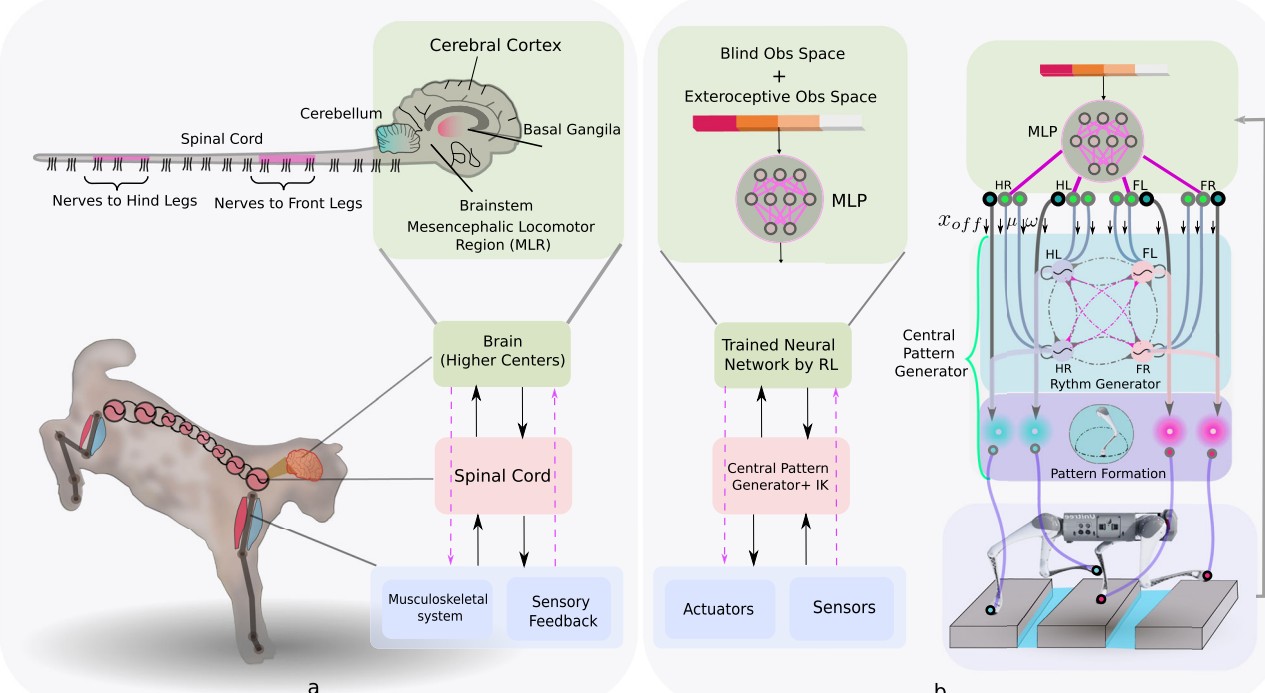

**Fig. 1 | Our proposed learning architecture. a** To model locomotion control, we consider three main interacting layers: the brain (higher centers), the spinal cord, and the body and sensory feedback modules. Higher neural centers (such as the brainstem, basal ganglia, cerebellum, and motor cortex) send descending drive signals to modulate the spinal circuits, and/or directly interact with lower modules (body and sensory feedback). **b** We represent these higher neural centers with a multilayer perceptron (MLP) with three hidden layers of [512, 256, 128] neurons

with `elu` activation. To represent the Central Pattern Generator in the spinal cord, we use nonlinear amplitude-controlled phase oscillators for modeling the Rhythm Generator (RG) layer, whose outputs are mapped to foot positions and then motor commands with inverse kinematics (IK) through a Pattern Formation (PF) layer. FR, FL, HR, and HL stand for front right, front left, hind right, and hind left, respectively. The brain figure in (**a**) has been re-drawn with inspiration from Grillner et al.[1] (CC BY 4.0).

(*Hydromys chrysogaster*), Virginia opossum (*Didelphis virginiana*), and tufted capuchin (*Sapajus apella*). We use second-order polynomial fitting for the CoT as used in ref. 6. The speed at which the CoT curve for a walking gait intersects with the CoT curve for a trotting gait represents the energetically optimal transition speed (EOTS). The EOTS can be found by extrapolating the walking and trotting curves, however there are some cases for which the intersection does not exist, such as for the domestic dog (Fig. 2b). Moreover, there is limited data available for the Australian water rat, so the extrapolation/interpolation of the fitted curve may not provide definitive information. For both the robot and the animals, switching gaits at the EOTS leads to a reduction in energy expenditure. However, it is important to note that the preferred transition speed does not align exactly with the EOTS for most animals. The domestic dog (b) and Virginia opossum (d) have the most similar CoT curves to the robot. For the robot and all animal species besides the Australian water rat, the CV of the stride cycle duration is reduced by a change in gait. Moreover, in the Virginia opossum (d) and tufted capuchin (e), the CV of the stride cycle duration has a peak near the gait-transition speed. It is noteworthy that training two separate neural network policies for walk and trot gaits allows us to extend the gaits to velocities beyond the EOTS. This is exemplified in the reduction of CV in the walk gait after the transition speed, a phenomenon not observed in animal data where, after the transition speed, there are no longer any walk gaits possible or available for the animals. Furthermore, we acknowledge that detailed behaviors may differ between animals and robots due to distinct body elements and mechanics. However, the substantial reduction in the CV of stride duration by changing the gait around the energetically transition speed for the robot demonstrates consistency with animal data.

Since the concept of viability is difficult to quantify, we investigated three additional quantities which we believe allow one to estimate whether one gait is more "viable" than another (Fig. 2B). The first quantity is the lateral Divergent Component of Motion (DCM) offset. Increasing the DCM generally corresponds to a more dynamic gait, which in turn increases the risk of falling. We observe that, as with the CV of the stride duration, switching gaits reduces the lateral DCM offset (panel (f)). The desired lateral DCM offset is zero, since the robot is walking forward in a straight line. The second quantity is the maximal lateral force that the robot can withstand before falling (panel (g)). Here again we see that the switch from walk to trot allows the robot to withstand higher forces. The third quantity is the average body angular velocity (panel (h)). Here we see that switching gaits decreases the average body angular velocity (panel (h)), improving stability. For all figures except (g), we report average values testing the policies for 35,000 samples (7 tests of 5 s of locomotion) since the standard deviations are small (i.e., less than 10% of the mean), and the standard deviations are reported in the supplementary data. For panel (g), the policies were tested for 350,000 samples, equivalent to an average of 50 tests of 7 s of locomotion each. In summary, our results show that on flat terrain, the transition from walking to trotting at a certain speed is not only useful to reduce the CoT, but also to increase the viability of the gait, making it more robust against lateral pushes, more periodic, with less angular velocity, and lower lateral DCM offset. Despite the enhanced viability achieved through the walk-trot transition, a comparison of panels (f) and (g) reveals that, for the trot gait, when the speed exceeds $1.2\,\mathrm{m\,s^{-1}}$ ($4.32\,\mathrm{km\,h^{-1}}$), the maximum allowable push before falling increases with an increasing DCM offset. This contradicts our viability analysis (please see the Supplementary Methods 3 and 4) based on the DCM offset. However, this outcome is somewhat anticipated, as the DCM offset is based on the linear inverted pendulum model (LIPM), which relies on assumptions such as zero centroidal angular momentum, constant body height, and sufficient friction.

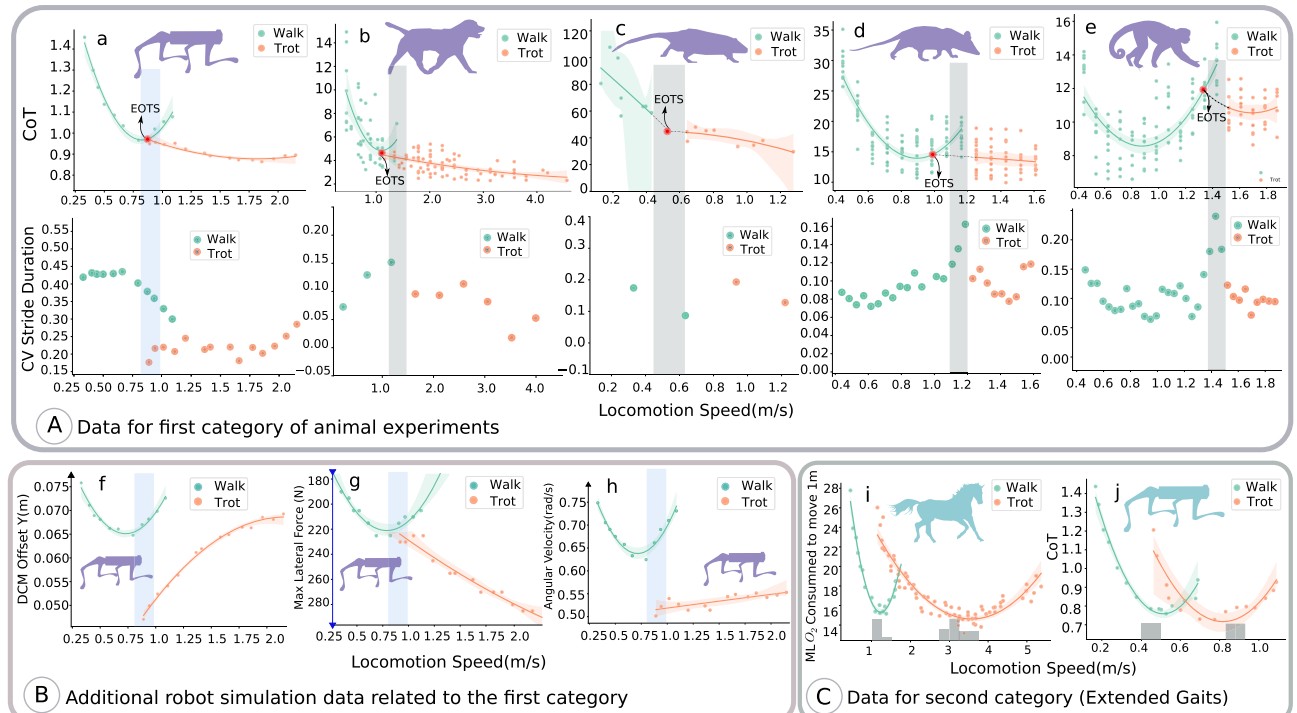

**Fig. 2 | Qualitative comparison data for robot and animal locomotion.** The CoT and CV of stride duration of the animals and the robot are plotted against the locomotion speed of walk-trot gaits in: (**a**) the quadruped robot, (**b**) the *domestic dog*, (**c**) the *Australian water rat*, (**d**) the *Virginia opossum* and (**e**) the *tufted capuchin*. This data relates to the first category of animal experiments with normal gaits from ref. 6. The shadow vertical bars in (**b**–**e**) represent the range of preferred gait transition speeds for animals. In (**a**) and (**f**–**h**), the blue bar indicates the expected transition speed. The speed at which the CoT curve for a walking gait intersects with the CoT curve for a trotting gait represents the energetically optimal transition speed (EOTS). **f** The lateral DCM offset, (**g**) maximum allowable external push, (**h**) average body angular velocity from the robot simulation in (**a**). **i** The CoT is plotted against locomotion speed for the second category of animal data, i.e., horses with extended gaits[2]. The shadow bar in (**i**) represents the speed at which horses prefer to locomote for that particular gait. In a lifelong learning process, this is the velocity for which the horse's motor control has been optimized for each gait[2]. **j** Robot data for extended gait. In (**j**), the gray shadow bar represents the speed at which the policy is trained. The shaded area around the interpolated curves indicates the 95% confidence interval of interpolation. For the CV of stride duration for the animal data, ref. 6 has calculated the mean of the stride duration for the specific speed interval. For external pushes in (**g**), we report the maximum lateral external force values for which the robot has a 100% success rate. The vertical axis of (**g**) is reversed for easier comparison with (**f**).

These assumptions do not hold for high-speed velocities, where limb rotation occurs rapidly.

In the second scenario of locomotion on flat terrain, we investigate the data of extended gaits from Hoyt and Taylor[2]. Horses continue to train their motor control through a lifelong learning process to locomote at certain limited speed ranges for each of their gaits[2]. To investigate locomotion principles, horses were briefly trained to extend their gaits to speed ranges in which they would not normally use that gait. During this process, horses learn to increase their velocity by lengthening their stride while maintaining a relatively constant frequency. While horses learn to optimize their motor control policies over a long period of time, they were taught to extent their gaits during a short period, experiencing new locomotion parameters which they had not previously encountered in their lifetime. In the context of robot locomotion, extending a gait can be understood as a scenario in which the robot is forced to locomote at speeds outside of the range optimized for during the training process. Figure 2C illustrates the results from the second experimental category: extended walking and trotting gaits for both the robot and horse. Interestingly, the resulting CoT curve shape for the robot is very similar to the data from the horses. Comparing panels (a) and (j) reveals that an extended trot produces a CoT-velocity curve with higher curvature. In other words, the CoT increases rapidly for speeds that are farther from those for which the robot was optimized for (and similarly for the horse, speeds that are farther from those it would choose naturally before the rapid training of extended gaits). Moreover, because we changed the stride length manually in panel (j), the policy experiences states for which it

had not been optimized for during locomotion. Thus, the viable range of speeds for panel (j) is smaller than for panel (a). Please refer to the Supplementary Method 5 for further investigation of the role of RL in flat terrain locomotion.

**Emergence of gait transitions in gap crossing scenarios**
We trained the robot to cross 8 challenging gaps in a row with 14 cm platforms between each gap in the PyBullet simulator (Fig. 3a). The gaps were randomized in width in the range of [14,20] cm. As exteroceptive sensory information, we include the distances between each foot and the front and back of the next upcoming gap in the observation space. The base and feet trajectories are shown in Fig. 3b. We observe that the agent has learned to increase the stride length, maximum feet height, and average body height in order to cross the gaps. Interestingly, Fig. 3b shows that the robot places its hind limbs where the front limbs were located in the previous stride, similarly to how cats place their rear paws in the pawprints made by their front paws (also known as direct registering)[40]. In order to ensure a reliable contact surface near the gaps, the robot places its front and hind feet in the same positions once it reaches the gaps. This strategy improves the predictability of the gait, which facilitates the anticipation of the non-viable states and thus increases gap traversing abilities. As shown in Fig. 3b, the Hind Left (HL) foot trajectory is higher than that of the Hind Right (HR) foot in the XZ plane. Such an observation can be attributed to the fact that we do not constrain the locomotion policy to have symmetry in the system, and this is due to increased body roll movement after starting to cross the gaps (Supplementary Fig. 7). Figure 3c

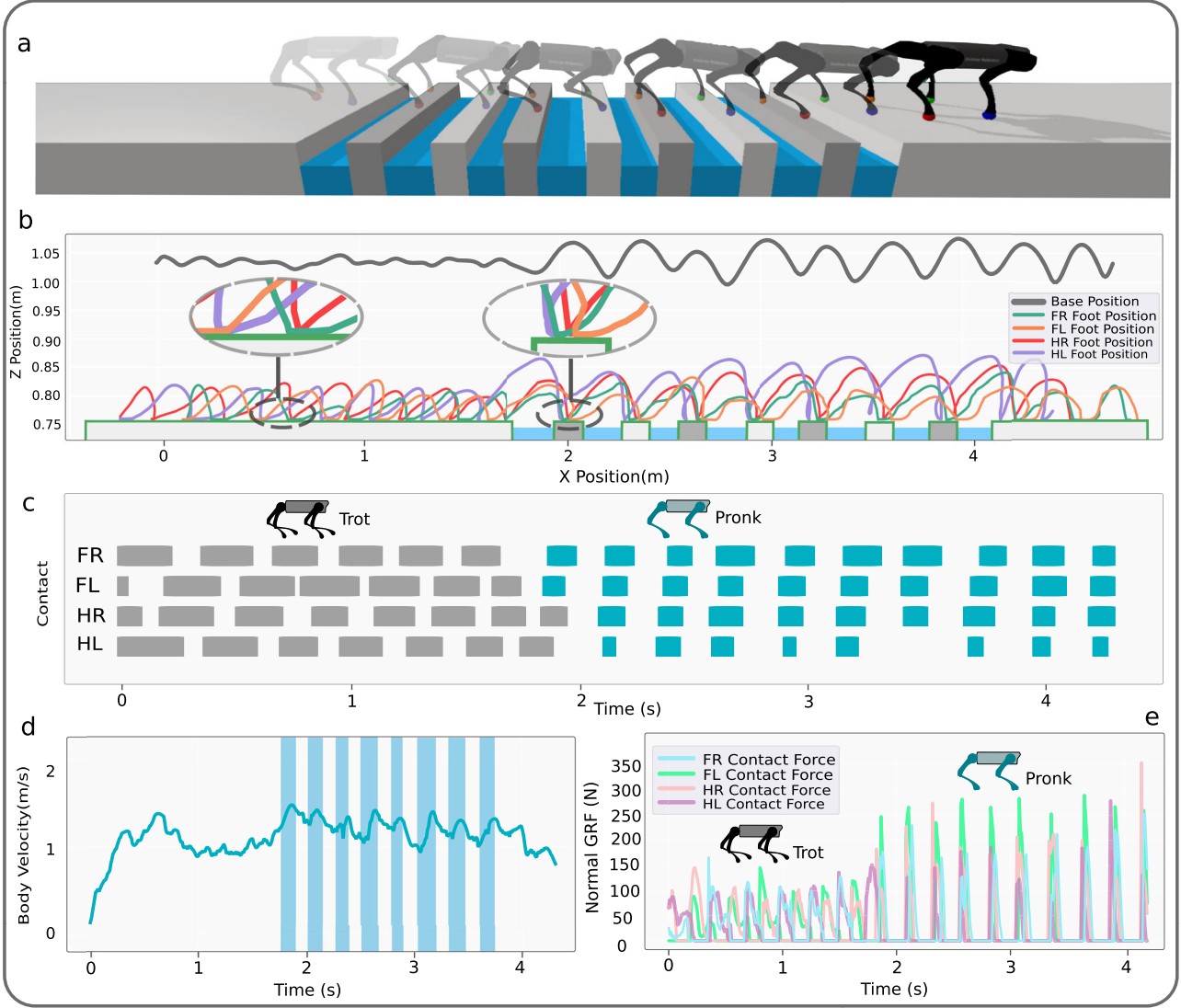

**Fig. 3 | Emergence of gait transition in the gap crossing scenario.** Crossing 8 gaps with randomized lengths between [14,20] cm, with only 14 cm contact surfaces. **a** Simulation snapshots. **b** Body position and foot positions in the XZ plane. **c** Foot contact duration, where blue represents contacts while crossing the gaps. **d** Body velocity in the longitudinal direction. The shadow bars indicate when the base is over a gap. **e** Normal contact (ground reaction) forces for each limb. FR, FL, HR, and HL stand for front right, front left, hind right, and hind left, respectively. The shadow bars in (**d**) indicate the duration during which the body center is crossing a gap (Supplementary Movie 2).

illustrates the duration that each foot is in contact with the ground. We observe a trotting gait prior to reaching the gaps, where diagonally opposite feet (e.g., front right and hind left) are in phase. However when crossing the gaps, the supraspinal drive has learned to transition gaits from trot to pronk, so that all feet are in the same phase (please see Supplementary Movie 2 for visual reference of these experiments). As shown in Fig. 3d, as the robot starts to cross the gaps, the supraspinal drive increases the robot's velocity. While we limit the reward the agent can receive by setting the desired forward velocity to 1 m s⁻¹, the agent has learned to increase the velocity of the robot to up to 1.5 m s⁻¹ to cross the gaps. It is noteworthy that in this scenario, changes in speed do not induce the transition; rather, the gait transition results in a change of speed. Table 1 shows the CV of the stride duration and length, as well as the CoT before and after reaching the gaps and the gait transition. There is a reduction in the CV of the stride duration and stride length after the gait transition, indicating a reduction of the kinematic and dynamic variability. However, the CoT increases after the gait transition, indicating an increase in energy expenditure for the pronk gait. Figure 3e shows the normal contact

forces for each limb, and in particular the increase of the peak contact forces after the trot-pronk gait transition. These results suggest that, in our experiments, energy efficiency and peak forces do not explain the gait transition, and that it is rather the avoidance of non-viable states that triggers the gait transition, as could be expected from this particular environment. Please refer to the Supplementary Method 7 for details regarding the supraspinal signals.

To analyze the impact of each component of the reward function, we trained 64 policies in Isaac Gym, considering four different values (high, medium, low, and zero) for the reward weights associated with viability, CoT, and peak contact forces

**Table 1 | The CV of the stride duration, stride length, and CoT before and after a gait transition (Fig. 3)**

|                       | CV Stride Duration | CV Stride Length | CoT  |
| --------------------- | ------------------ | ---------------- | ---- |
| Before gait transition | 0.38               | 0.44             | 0.84 |
| After gait transition  | 0.31               | 0.35             | 0.93 |

(please see Supplementary Movie 3 for visual reference of these experiments). Figure 4 shows the success rate, CoT, and deviation of the contact force from its maximum value threshold across various cases. As exteroceptive sensory information, we include a height-map of the terrain in front of the robot, beginning below its front hips (see Supplementary Fig. 1). Figure 4a shows the weights of each case. The highest success rate is obtained with case 54, in which viability, CoT, and peak forces are assigned high, low, and low weights, respectively, as in the PyBullet results. It is noteworthy that this case exhibits the highest success rate of 99.25% across 2400 gap attempts, and CoT and peak forces increase after the gait transition from trotting on flat terrain to pronking over the discrete gap terrain.

Case 33, with a medium weight for viability and zero weights for CoT and peak forces, achieves the second highest success rate of 98.67%. However, the CoT and peak contact forces exhibit a deterioration after the transition. Among cases 34, 37, 51, which all have a success rate above 98%, it is observed that in two cases the CoT decreases after the gait transition, while in case 37 it remains constant. The peak forces increase after the gait transition in all three of these cases. This suggests that the improvement of CoT and peak forces is not necessarily guaranteed after the gait transition. In cases 34 and 51, where the CoT decreases after the transition, it is noteworthy that the weight assigned to the CoT minimization in the reward function is zero.

For cases 1–16 which have zero viability weight, the robot remains stationary for the whole episode and does not exhibit any movement. Likewise, in cases 17 to 32 with a low weight for viability, all instances, except for 17 and 18, result in the robot remaining stationary. In cases 17 and 18, the robot learns to walk with a low reward weight for viability, zero weight for energy efficiency, and either low or zero weight for the contact force term. However, it comes to a halt before reaching the brink of the gaps, and does not cross them.

We observe an increase in peak forces after the transition in all cases which have a success rate higher than 50%. In case 36, with high

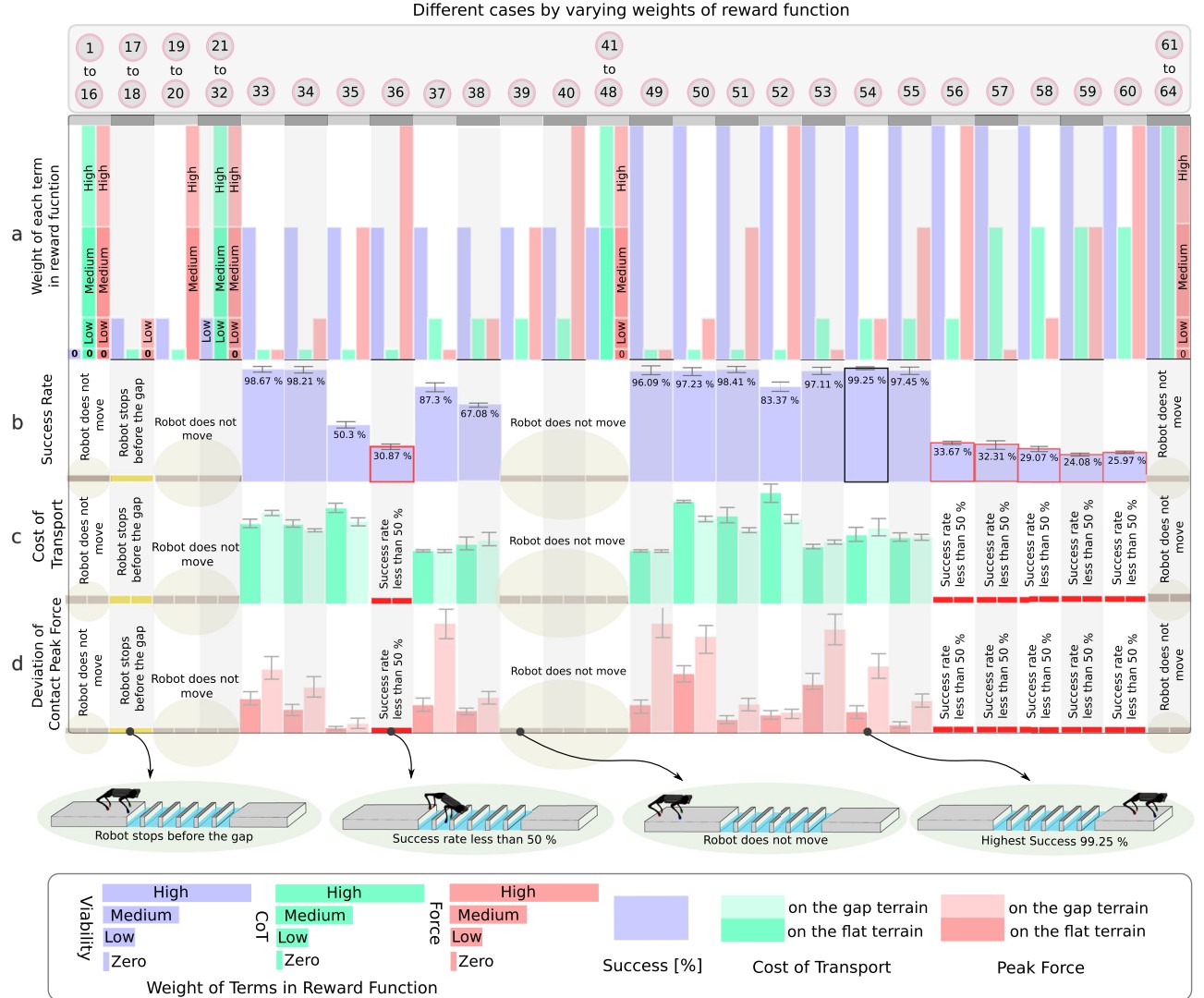

**Fig. 4 | Reward function analysis.** Quantitative results from testing 64 policies with varying weights in the reward function for crossing two series of 6 consecutive gaps (Supplementary Movie 3). We consider four values of zero, low, medium, and high for the weights of the reward function terms for viability, Cost of Transport (CoT), and peak contact forces in (**a**). The zero, low, and medium weights are selected to be approximately zero, 10%, and 50% of the high values. We report average (**b**) success rate, (**c**) CoT, and (**d**) peak contact forces for testing the polices for 4400 samples each (200 attempts of crossing 12 gaps). We show mean values and the standard deviations for success rate and CoT. Cases demonstrating similar behavior, like cases 1 to 16, have been grouped in the same column. To evaluate the peak forces, we first separately trained locomotion controllers on flat terrain to reach a velocity of 1.2 m s$^{-1}$, which is the average gap crossing velocity, and observed peak contact forces of 180 N. For the gap crossing scenario, our reward function penalizes peak contact forces above this threshold at each control cycle, and the plots show the mean contact force in excess of 180 N across all tests. The error bars indicate the standard deviation.

weights for viability and peak contact forces, we observe that the robot learns to walk on flat terrain; however, it achieves a success rate of less than 50%. From case 56 onwards, where either high weights are assigned to contact forces or medium to high weights are assigned to CoT, a significant decrease in success rate is observed compared to case 55, which has low and medium weights for CoT and peak contact forces respectively.

## The role of sensory features in gait transitions

As shown in Fig. 5, we investigate the effects of observing different combinations of exteroceptive, proprioceptive and vestibular sensory features for gap crossing scenarios on criteria such as success rate, average angular body velocity, the CV of the stride duration, the CV of the stride length, the CoT, and the Average Lateral DCM offset (please see Supplementary Movie 4 for visual reference of these experiments). In particular, we study which sensory feature combinations are necessary and sufficient to learn to successfully cross variable gaps, and we analyze the effects through ablation experiments in cases 1–10. We also investigate the effects of adding explicit oscillator coupling between oscillators such that the agent must walk, trot, or bound in cases 11–13. For the purposes of evaluation, we perform 14 policy rollouts, and present average results across all tests.

Figure 5a shows the success rate as the bar height, while the color indicates the average body angular velocity. Our results show the highest success rates for policies 1, 2, and 3, which observe (1) the distances of all feet to the gap, (2) the distances only of the front feet to the gap, and (3) LiDAR depth measurements in front of the robot, respectively, in the observation space. The average body angular velocities of these three policies show that they also generate the smoothest gaits. Cases 4 and 5 show that removing contact force booleans and proprioceptive information from the observation space reduces the success rate by approximately 20%. The removal of front-feet distances to the gap in case 6 leads to a 50% reduction in the success rate. Case 7 shows that removing vestibular information (IMU data) from the "blind" sensory information leads to only a 16% success rate. Case 8 shows that sensing the base distance to the gap, rather than the explicit feet distances to the gap, results in only a 23% success rate. The lowest success rates are for cases 9 and 10, where only instantaneous contact/penetration into a gap, or no gap information at

all are included in the observation space. Finally, in cases 11–13, we observe that phase couplings of walking, trotting, and bounding gaits lead to success rates of 48%, 13%, and 25%, respectively, indicating that such couplings (which impose a particular gait) have a detrimental effect on the success rate. This suggests that supraspinal drive might play an important role in modulating coupling strengths for anticipatory locomotion.

Figure 5b shows the CV of the stride duration (bar height) and the CV of the stride length (color). Of the cases with a high success rate, case 1, which includes foot distance to the gap in the observation space, has the lowest CV of stride duration. The CV of the stride duration does not vary significantly between the other cases with high success rates. Cases 1 and 2, with explicit visually-extracted information of feet distances to the gaps, have the lowest CV for the stride length.

Figure 5c shows the CoT (bar height) and average lateral Y DCM offset (color). Of the cases with high success rates (cases 1–3), cases 1 and 2 show that having all/front feet distances to the gap as explicit exteroceptive sensing leads to a lower CoT. The CoTs of the other cases are not as informative, as they have low success rates where the robot frequently falls into a gap. The lowest average lateral DCM offsets are for cases 1 and 2, which include all-feet distances, or only front-feet distances, to the gap in the observation space. These results suggest that front-feet distances to the gap are necessary and sufficient explicit sensory features to avoid non-viable states (i.e., falling into a gap).

## Hardware experimental results

Figure 6a shows snapshots of a sim-to-real transfer to the A1 hardware of a policy trained with our proposed method for a task of crossing four consecutive gaps with widths of 30 cm, 21 cm, 18 cm, 14 cm. We simplify the sim-to-real transfer by using knowledge of the relative gap distances to the robot from an equivalent scenario completed in simulation, instead of using on-board vision or LiDAR. The robot crosses this challenging gap scenario with a velocity of 1.3 m/s, and we observe a trot-pronk gait transition when reaching the gaps, and a pronk-trot gait transition after crossing the last gap (please see the Supplementary Movie 1).

Table 2 compares the performance of the proposed bio-inspired controller with other quadruped robot controllers for gap-crossing

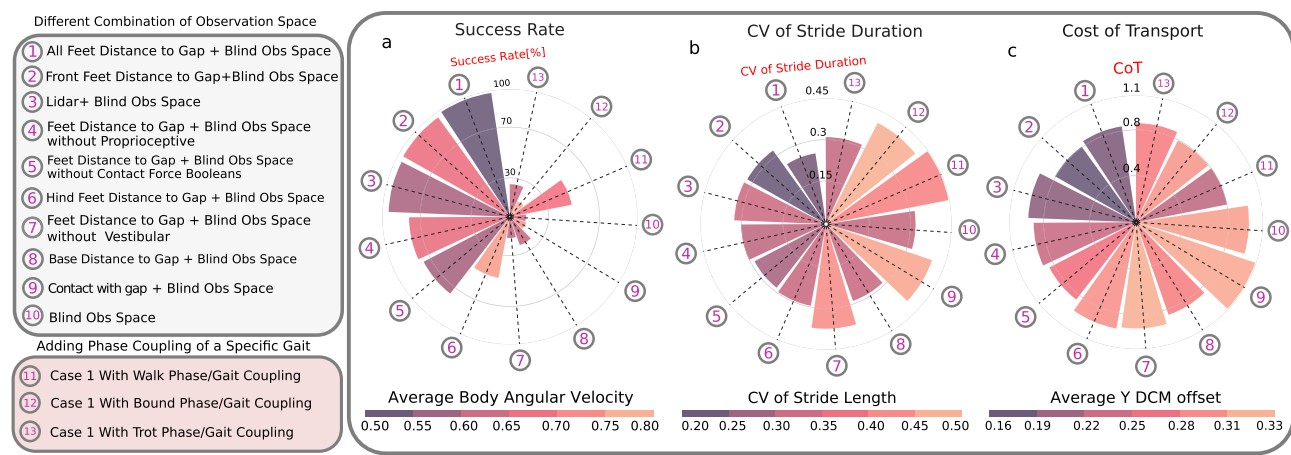

**Fig. 5 | Observation space and phase coupling analysis.** Quantitative results from testing 13 policies with varying observation spaces and phase coupling for crossing a series of four consecutive gaps (Supplementary Movie 4). We report: (**a**) average success rate, body angular velocity, (**b**) coefficient of variation (CV) of stride duration/length, (**c**) Cost of Transport (CoT), and lateral divergent component of motion (DCM) offset for testing the polices for 50,000 samples each (14 attempts of crossing four gaps). Policies 1–10 are trained using a variety of combinations of "blind" (non-visual) and exteroceptive visual sensing. We only show mean values

since the standard deviations are small (i.e., less than 20% of the means). The standard deviations are reported in the supplementary data. Policies 11–13 are trained with oscillator couplings to force walking, trotting, and bounding gaits, respectively. Exteroceptive sensory feedback features include LiDAR measurements, as well as geometrically-extracted quantities such as feet distances to the gap, base distance to the gap, and foot contact/penetration into a gap (visualized in Fig. 7a).

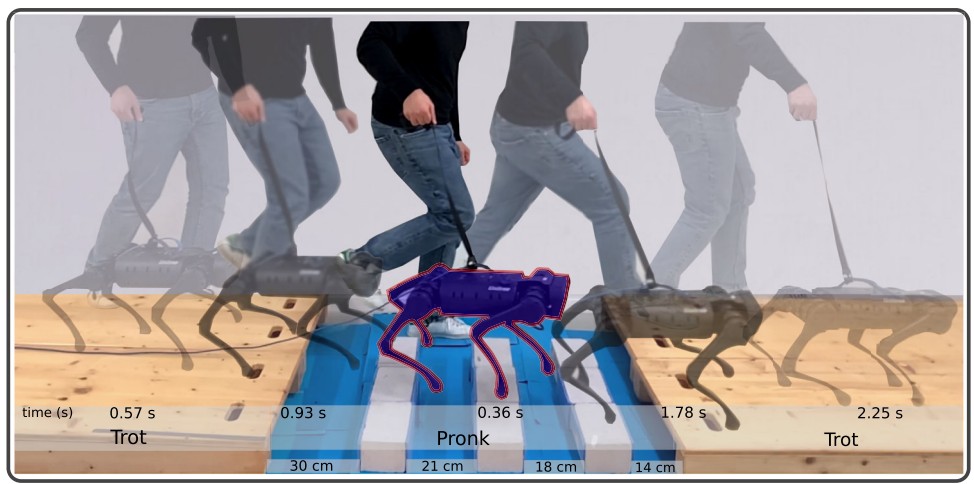

**Fig. 6 | Experiment for gap crossing scenario.** Crossing four gaps with widths of 30 cm, 21 cm, 18 cm, and 14 cm with a velocity of 1.3 m/s. We observe the emergence of a trot-pronk gait transition when approaching the gaps, followed by a pronk-trot gait transition after successfully crossing the last gap (Supplementary Movie 1).

**Table 2 | Comparison of the DeepTransition gap-crossing controller with state-of-the-art methods (hardware results)**

| Research | Robot | Gap | Max Gap width / Body length | Beam width | Speed | Froude | Controller |
|---|---|---|---|---|---|---|---|
| Kalakrishnan et al.[61] | Little Dog | Single gap | 0.772 | N/A | 0.385 | 0.007[b] | Optimization |
| Magana et al.[62] | HYQ | Consecutive[a] | 0.135 | 20 cm | 1.8 | 0.046 | Optimization |
| Yu et al.[63] | Laikago | Single gap | 0.4 | N/A | 1.35 | 0.032 | MPC-RL |
| Lee et al.[64] | Laikago | Single gap | 0.4 | N/A | 1.35 | 0.032 | MPC-RL |
| Agarwal et al.[65] | Unitree A1 | Consecutive[a] | 0.72 | 30 cm[b] | 1.26 | 0.041 | RL-Supervised |
| Xie et al.[66] | Unitree A1 | Single gap | 0.41[b] | N/A | 1.08 | 0.03[b] | RL-Optimization |
| Agrawal et al.[67] | Unitree A1 | Consecutive[a] | 0.5 | 20 cm | 0.9 | 0.021 | Optimization |
| Margolis et al.[68] | MIT-Cheetah | Single gap | 0.4 | N/A | 4.5[a] | 0.549[a] | MPC-RL |
| Rudin et al.[69] | ANYmal C | Single gap | 0.9[a,b] | N/A | 3.6[b] | 0.203[b] | RL-Curriculum |
| Yang et al.[70] | Unitree A1 | Single gap | 0.83[a] | N/A | 1.44 | 0.054 | RL-Supervised |
| DeepTransition[a] (Ours) | Unitree A1 | Consecutive[a] | 0.83[a] | 14 cm[a] | 4.68[a] | 0.574[a] | CPG-RL |

Speeds are expressed in km h$^{-1}$. Consecutive gaps corresponds to scenarios where the robot crosses multiple gaps in a row with small distances between gaps (the distance between gaps is less than the body length). The beam width indicates the distance between the gaps. We do not report beam width for the single gap scenario since there is considerable distance (more than the body length) between gaps. The Froude number ($v^2/(g \cdot h)$) is a dimensionless number that is useful for size-independent comparisons of animal and robot agility. (g), (v) and (h) are gravity acceleration, forward velocity, and nominal base height respectively.
[a]represent the best outcomes for each respective parameter.
[b]denotes an estimation from the corresponding publication video.

scenarios. The proposed controller outperforms the previous state-of-the-art controllers by showing the ability to cross the most challenging consecutive gaps of up to 30 cm (0.83 gap/body length ratio), with only 14 cm beam contact widths between gaps. Furthermore, our hierarchical biology-inspired policy generates the most agile gap crossing with a velocity of 1.3 m/s. These results suggest that a better understanding of animal behavior through testing biological hypotheses with robots can help improve robot locomotion performance.

## Discussion

We began this study by evaluating the consistency of the locomotion policies learned with our framework with available animal data for walking and trotting on flat terrain. The CoT and CV of the stride duration were found to be qualitatively consistent across both considered categories of animal experiments and robot simulations, namely walking and trotting with and without extended gaits. Furthermore, we investigated why the CoT-velocity curve for trotting has a higher curvature for the second category of animal experiments, concluding that the difference is mainly attributable to the extension of gaits in these experiments. We simulated extended gaits by varying the stride length, taking inspiration from previous horse experiments[41]. More specifically, once we had trained walking and trotting gaits at particular velocities, we manually altered the stride length mapping parameter in order to achieve locomotion at velocities outside of those trained for with the same gait. We observed that the CoT of the extended gaits increased rapidly as the robot velocity moved away from its optimal point for both robots and horses. For this reason, the curvature is greater in this case than in the first category of data, which did not include extended gaits. We hypothesize that the large increase in CoT in relation to the changing velocity can be explained by the fact that the locomotion policies of both the animals and robots have not been optimized for these parameters. Overall, our study of flat terrain locomotion tends to show that for both animals and for our quadruped robot, it is worth switching from walk to trot when increasing speed. The switch is beneficial both for increasing the energy efficiency of the gait (lowering the CoT) and for increasing the viability of the gait (e.g., increasing the robustness against lateral pushes). Our findings suggest that a quadruped robot can be a useful tool for testing biological hypotheses about quadruped animals.

In the second part of this study, we tested whether viability can be a determinant of gait transitions in quadruped locomotion, concluding that gait transitions can be triggered by environment perception to

avoid unviable states in challenging terrain. We can make the following observations:

- In situations where the robot was constrained to locomote by walking, trotting, or bounding gaits, it was unable to learn to solve challenging gap crossing scenarios with a high success rate. This observation suggests that the trot-pronk transition emerges to prevent the robot from falling into gaps (i.e., to prevent non-viable states). Note that the pronk gait was here particularly well-suited for the specific gaps that we used, and that other gaits could likely have emerged for other types of gap distributions (in particular if left and right legs would have faced different gap widths and placements).

- Our systematic analysis of the reward function term weights for viability, CoT, and peak forces demonstrates that viability is the most critical term to accomplish successful gap crossing, and high weights for penalizing the CoT and/or peak forces results in an inability to learn gap crossing abilities. This unsurprising finding highlights the significance of viability in the locomotion task.

- Energy efficiency and peak forces have not necessarily been improved after the trot-pronk gait transition, which shows that, in our case, a gait transition is not triggered in order to reduce energy expenditure, and while the pronk gait can serve as a way to cross gaps, it may also increase the risk of injury to the body (musculoskeletal system). The CV of the stride length is reduced by the transition from trot to pronk, which shows that following the gait transition, the predictability of the gait increases.

- From all tested exteroceptive sensory features, the highest gap crossing success rate was found for policies which included feet distances to the gap in the observation space. This information can be directly used by the policy to forecast and anticipate planning future footstep locations. In particular, we observe that front-feet distances to a gap are the most important exteroceptive sensory features, which are both necessary and sufficient explicit features for learning to cross gaps.

- If the only exteroceptive sensing the agent has access to are the front feet distances to a gap, this forces the agent to learn an internal kinematic model by combining front foot position, internal CPG states, and proprioceptive sensing to modulate the "blind" hind leg motions to successfully cross the gaps. This result corroborates the hypothesis that cats and horses control their front legs for obstacle avoidance, and the observation that their hind legs follow the previous support location of the front legs based on internal kinematic memory[40,42]. This is also known as direct registering. Similarly, our robot places its hind limbs approximately where the front limbs were located in the previous stride, which enhances gait predictability by lowering the CV of the stride length. However, it is worth noting that the CV of the stride length must increase when crossing irregular gaps.

- Our results suggest that vestibular feedback has a high impact on viability in the emergence of trot-pronk gait transitions in our designed gap crossing scenarios. A previous study suggests that vestibular feedback has the greatest impact on the landing behavior of cane toads[43]. In our study, the landing phase is the most challenging phase of the emerged pronk gait as the agent must control and plan footholds and body orientation to avoid falling into a gap, and our results support that vestibular sensing has a high impact on successful landing. For example, Case 7 of Fig. 5 shows a drastically reduced success rate when removing vestibular information from the observation space.

- Finally, we demonstrate that our hierarchical biology-inspired control architecture (Fig. 7) enables the Unitree A1 quadruped robot to cross challenging gap terrains of up to 30 cm in width (83.3% of the body-length) in sim-to-real hardware experiments. To the best of our knowledge, this represents the most dynamic crossing of such large consecutive gaps for a quadrupedal robot, where A1 exhibits a trot-pronk gait transition to locomote at over 4.68 km h$^{-1}$ (1.3 m s$^{-1}$). Moreover, this is the first learning-based locomotion framework in which gait transitions emerge

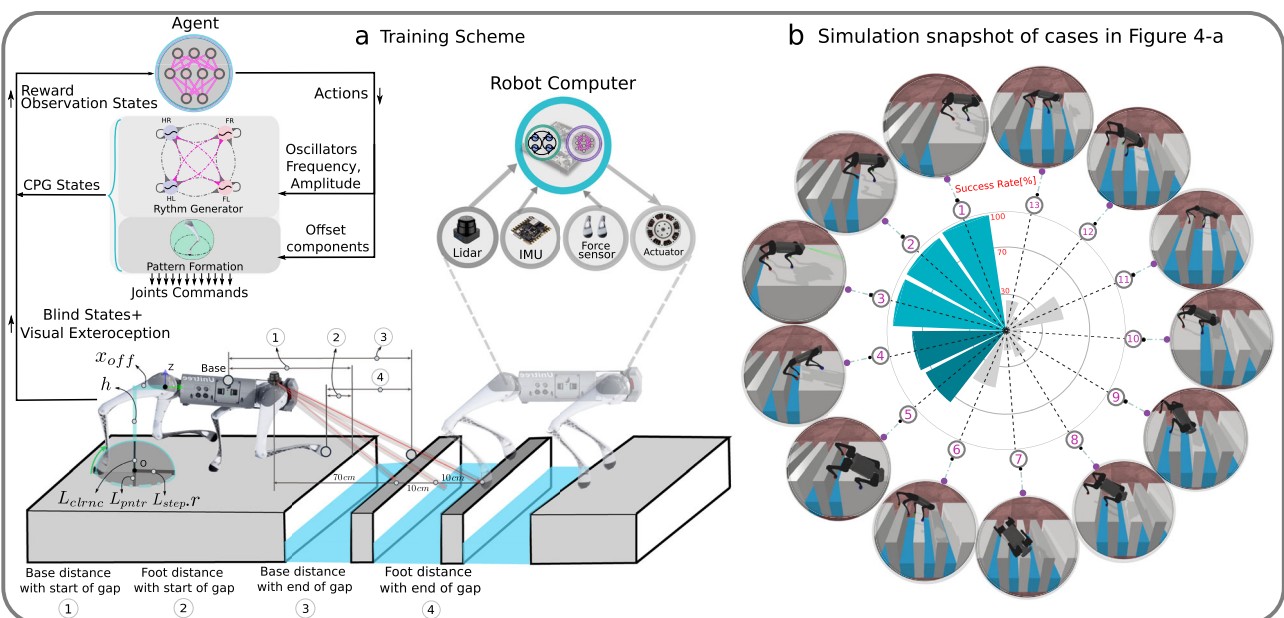

**Fig. 7 | Training scheme. a** Training scheme and schematic visualization of feet trajectory and visual exteroceptive feedback features. The oscillatory trajectory is built around a central point $O$. The offsets $x_{off}$ are used to change the central point of oscillation. $x_{off}$ is a horizontal offset between the set point of oscillation and the center of the hip coordinate, controlled directly by the supraspinal drive, bypassing the CPG dynamics. $L_{step}r$ is the step length multiplied by the oscillator amplitude, $h$ is the nominal leg length, $L_{clrnc}$ is the max ground clearance during leg swing phase, and $L_{pntr}$ is the max ground penetration during stance. The desired foot positions are mapped to motor commands and tracked with joint PD control, and sensing includes LiDAR for visual perception, an Inertial Measurement Unit (IMU) to filter base velocities and orientation, and foot contact sensors for measuring contact forces. **b** Simulation snapshots of cases of Fig. 5a.

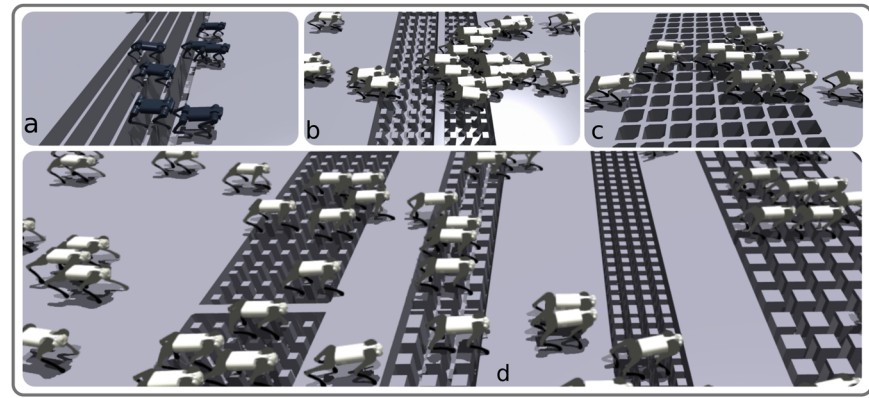

**Fig. 8 | Training policy for different discrete terrain.** Training gap-crossing locomotion policies with different robots (Unitree A1, (**a**) and Go1, (**b**–**d**)) on various discrete terrains in Isaac Gym (Supplementary Movies 5–8).

spontaneously during the learning process without having a dynamical model, MPC, curriculum, or mentor in the loop. Moreover, Fig. 8a–d illustrates the potential of the controller for training locomotion on various discrete terrains (please see Supplementary Movies 5–8 for visual reference of these experiments).

Several previous biological studies have suggested that gait stability can be considered as a primary determining factor for gait transitions of quadruped animals on flat terrain[6,28,44]. These studies suggested that CoT may be used as a surrogate for gait stability or as a secondary objective. In this paper, we propose viability as a comprehensive criterion for gait transitions, generalizing the concept of gait stability. In particular, our simulation results in the anticipatory scenario propose that viability is the primary objective for the emergence of gait transitions on variable discrete terrains. These observations suggest viability could be the universal and primary objective of gait transitions, while other criteria are secondary objectives and/or a surrogate of viability.

We believe that this paper represents a useful starting point for using robots and deep learning to investigate the determinants and triggers of gait transitions. There is, however, much room for further investigation. For example, the nonlinear oscillators we used are only first-order approximations of CPG circuits and are useful for investigating neuroscience research questions at a high level. However, a more detailed understanding of the underlying circuits (for example for identifying the descending pathways that contribute to gait transitions) would require more detailed and genetically-identified models for CPG neural circuits[45].

The dynamics of the musculoskeletal system also play a key role in the robustness and energy efficiency of animal locomotion. We therefore plan to incorporate a simulation of the animal musculoskeletal system (for example with pairs of antagonist muscles, as well as biarticular muscle models) and investigate its role for gait transitions. As mentioned earlier, there is evidence that gait transitions may occur to reduce the mechanical load on the musculoskeletal system and joints[5]. We believe that our framework will provide a useful tool for investigating this problem, by directly penalizing the muscle peak forces and joint jerk in our reward function, and using these values as feedback to the CPG or as a part of the observation space. Additionally, it has been suggested that contact loading feedback to the CPG can trigger the gait transition by increasing locomotion frequency and speed on flat terrain[24,25]. By incorporating contact loading feedback to the CPG, the proposed architecture can be extended to investigate the interaction between supraspinal drive and loading feedback

for triggering gait transitions on flat terrain. In summary, the combination of robotics, simulation, and DRL will likely lead to a better understanding of key aspects of animal locomotion, as well as to more agile locomotion controllers for legged robots.

## Methods

### Central pattern generators

The locomotor system of vertebrates is organized such that the spinal CPGs are responsible for producing basic rhythmic patterns, while higher-level centers (i.e., the motor cortex, cerebellum, and basal ganglia) are responsible for modulating the resulting patterns according to environmental conditions[1]. Rybak et al.[46] have proposed that biological CPGs typically have a two-level functional organization, with a half-center rhythm generator (RG) that determines movement frequency, and pattern formation (PF) circuits that determine the exact shapes of muscle activation signals. Similar organizations have also been used in robotics, for example in refs. [47,48]. Here we reuse the controller presented in refs. [39,47].

**Rhythm Generator (RG) Layer.** We employ amplitude-controlled phase oscillators to model the RG layer of CPG circuits in the spinal cord, as they are able to modulate the output signal by changing a few decision variables[21]:

$$\dot{\theta}_i = \omega_i + \sum_j r_j w_{ij} \sin(\theta_j - \theta_i - \phi_{ij}) \tag{1}$$

$$\ddot{r}_i = \alpha\left(\frac{\alpha}{4}(\mu_i - r_i) - \dot{r}_i\right) \tag{2}$$

where $r_i$ is the amplitude of the oscillator, $\theta_i$ is the phase of the oscillator, $\mu_i$ and $\omega_i$ are the intrinsic amplitude and frequency, $\alpha$ is a positive constant representing the convergence factor. Couplings between oscillators are defined by the weights $w_{ij}$ and phase biases $\phi_{ij}$. We use one oscillator for each limb.

**Pattern Formation (PF) Layer.** To map from the RG states to joint commands, we first compute corresponding desired foot positions, and then calculate the desired joint positions with inverse kinematics. This represents the Pattern Formation (PF) layer, and the desired foot position coordinates are formed as follows:

$$x_{i,\text{foot}} = x_{\text{off},i} - L_{\text{step}}(r_i)\cos(\theta_i) \tag{3}$$

$$z_{i,\text{foot}} = \begin{cases} -h + L_{\text{clrnc}}\sin(\theta_i) & \text{if } \sin(\theta_i) > 0 \\ -h + L_{\text{pntr}}\sin(\theta_i) & \text{otherwise} \end{cases} \qquad (4)$$

where $L_{\text{step}}$ is the step length, $h$ is the nominal base height, $L_{\text{clrnc}}$ is the max ground clearance during swing, $L_{\text{pntr}}$ is the max ground penetration during stance, and $x_{\text{off}}$ is a set-point that changes the equilibrium point of oscillation in the $x$ direction. Modulating the foot horizontal offset $x_{\text{off}}$ represents direct supraspinal control of the general position of the limb, bypassing the rhythm generation layer. A visualization of the foot trajectory is shown in Fig. 7a.

## Locomotion metrics

**Viability.** In control theory, Lyapunov stability of an equilibrium point means that solutions starting within some distance from the point will remain "close enough" forever. Any perturbations will lead only to minor and transient changes for the state variables of a stable system. Mathematical tools for dynamical systems can be used to analyze gait transition stability[28], however they are limited to the internal dynamics of the system without considering environmental constraints. For example, a robot may have a stable gait (in the sense of Lyapunov stability), but still fall when it steps into an unanticipated gap. Legged locomotion, when trying to maintain balance and avoid such falls or collisions with obstacles, can be considered to be a problem of viability rather than Lyapunov stability[49]. If **F** is the set of states in which the system is considered to have fallen, then the viable states are the set of states from which the robot can avoid entering **F** (see Supplementary Fig. 2)[38]. In this article, we use viable states, rather than stable states, to fully capture the concept of fall prevention in legged locomotion. It is noteworthy that the basin of attraction and viability share similarities in concept; however, they also have their differences. In the topic of bipedal locomotion, the basin of attraction represents the set of initial conditions or states from which the system, when perturbed slightly, will converge to a stable walking gait or pattern[50]. Therefore, viability has a close relationship with the available control inputs (action), but the basin of attraction focuses on the initial conditions of the system. Moreover, viability takes into consideration environmental constraints[51].

On variable terrains with discrete footholds, a robot can fall by stepping into a gap, thereby entering a non-viable state. Unfortunately, in general the computation of the viability kernel can be intractable (except for the simplified pendulum model[52,53]), and due to the complexity of the multi-body dynamics of walking systems, it may be numerically extremely expensive or even impossible to check whether a given state is viable or not[54]. However, when learning gap crossing skills, it is possible to define the reward function such that it promotes not entering into non-viable states (stepping into a gap), and thus promotes viability. Viability can also be indirectly evaluated by considering the success rate of traversing variable environmental conditions, for example the number of gaps that the robot is able to successfully cross out of the total number of gaps encountered.

Griffin et al.[44] hypothesized that horses transition gaits from walking to trotting on flat terrain based on the stability of the inverted pendulum dynamics. On flat terrain, the viability of locomotion based on the LIPM approximation can be evaluated based on the DCM offset[55]. The DCM (also called Capture Point or Extrapolated CoM) dynamics is the unstable component of the LIPM, and the DCM offset (distance between the DCM and the Center of Pressure (CoP)) specifies the rate of divergence for the unstable component (see the Supplementary Method 3). The DCM offset becoming greater than a certain threshold will lead to entering non-viable states, and this threshold can be calculated for bipedal locomotion assuming a single support phase[56]. Although this does not apply to quadrupedal locomotion, we

can still use the DCM offset as an indicator of approaching the boundary of the viability kernel. In our experiments, the desired lateral motion is zero, and therefore an increase in the lateral DCM offset corresponds to a reduction in viability. Please refer to the Supplementary Methods 3 and 4 for a detailed analysis of viability using the LIPM, as well as the interconnection between viability, DCM offset, and robustness during locomotion on flat terrain. To explore the viability condition of the full robot and validate the DCM offset analysis, we conducted external push experiments to assess the maximum push that the robot can withstand before falling. For each locomotion speed corresponding to Fig. 2f, we conducted multiple experiments, varying the force from 190 N to 400 N. Each force setting was tested for 350,000 samples across approximately 50 episodes, each lasting 7 s. The success rate corresponds to the percentage of episodes where the robot completes the trial without falling. We then determined the largest force at which the robot achieved a 100% success rate.

**Stride duration variability (periodicity).** We propose that viability on flat terrain can also indirectly be evaluated by inter-stride variability. Indeed, locomotion gaits with high inter-stride variability increase the risk of inter-limb interference, making tripping and/or falling more likely on flat terrain. Therefore, variability in inter-stride parameters has important biomechanical consequences for gait stability[6,57]. This variability can be computed by the Coefficient of Variation (CV), which is defined as the ratio of the standard deviation to the mean of the data, and indicates the variability. We compute the CV of the stride duration for whole strides in one episode, and use this quantity to evaluate viability on flat terrain (but not on irregular terrain).

**Stride length variability (predictability).** The CV of the stride length is used to evaluate the predictability of a locomotion gait. A low stride length CV means that the stride length is relatively constant, and that the next step position can be predicted based on previous steps. Predictability of a gait therefore simplifies the anticipation of future feet placements. Similar to periodicity, predictability is advantageous in the case of regular and structured terrain. The CV is calculated for all whole strides by all limbs during a training episode, and the mean is then calculated over all episodes.

**Energy efficiency.** We compute energy efficiency for a system with the dimensionless Cost of Transport (CoT). The CoT formula is defined as $\text{CoT} = \frac{P}{m \cdot g \cdot v}$, where $P$ is the average power, $v$ is the average velocity, $m$ is the mass of the system, and $g$ is the gravitational acceleration.

**Gait smoothness.** To evaluate gait smoothness, we analyze the robot body oscillations during locomotion, and in particular the average angular velocity of the robot body $\bar{\omega}_{\text{Body}} = (\sum_{t=1}^{N} |\omega_{x,t}| + |\omega_{y,t}| + |\omega_{z,t}|)/(3N)$. High (absolute) angular velocities tend to correspond to shaky gait patterns.

**Peak force.** We evaluate peak force by measuring the peak ground reaction force. We use this as a proxy for representing peak forces that an animal would experience in its joints and muscles.

## Quantification of viability on flat and gap terrain
In this paper, viability is promoted in the reward function during the training of the locomotion policy, and then quantified and evaluated during test time using specific metrics. Here, we summarize and recap specific metrics used for viability quantification during test time, along with the reward function components for both flat terrain locomotion and gap crossing scenarios.

**Gap crossing.** In the reward function, we promote forward progress in a gap crossing scenario, which indicates that the robot is able to traverse the gaps without falling. The weights assigned to the reward

function are detailed in the Supplementary Method 2. We incorporate four different weight values: zero, low (10%), medium (50%), and high (100%). The assessment of viability in the gap-crossing scenario during test time involves quantifying the success rate of traversing variable environmental conditions. This can for instance be evaluated by determining the number of gaps the robot successfully crosses out of the total number of encountered gaps.

**Flat terrain.** In the reward function for learning locomotion on flat terrain, we have three terms that promote viability. Forward progress promotes viability of the system, as continuous high forward progress indicates that the robot has not fallen. Velocity tracking reward also promotes viability of the system, as tracking the steady state velocity indicates that the robot has not fallen. This term is already incorporated in the forward progress term; however, on flat terrain we would like to track different specific desired velocities to determine the relationship between the CoT and body velocity, as shown in Fig. 2. Large roll and pitch angles increase the likelihood of falling, thus penalizing body orientation helps to prevent the system from getting closer to the boundary of the viability kernel. For quantifying the viability on flat terrain during test time, we use the following metrics: (1) DCM offset, (2) Average Body Angular Velocity, (3) CV of the stride duration, and (4) maximum external lateral force (as shown in Fig. 2). It has been shown that by increasing the DCM offset, the system state approaches the boundary of the viability kernel, and therefore increases the risk of falling[56]. Granatosky et al.[6] propose that gait stability on flat terrain can also indirectly be evaluated by inter-stride variability. Indeed, locomotion gaits with high inter-stride variability increase the risk of inter-limb interference, making tripping and/or falling more likely on flat terrain. We also investigate the maximum external push that the robot can tolerate without falling at each specific speed. The Supplementary Fig. 3 shows the maximum external pushes for each specific speed of locomotion on flat terrain. The DCM offset and maximum external push exhibit reasonable correlations for locomotion speeds lower than 1.2 m s$^{-1}$ (4.32 km h$^{-1}$).

## Data from previously performed animal experiments
We assess the consistency of our learning architecture via a qualitative comparison with animal data on flat terrain, which involves plotting the CoT and the coefficient of variation (CV) of the stride duration against locomotion speed. We use two categories of animal data collected in previous studies.

**Normal gaits on flat terrain.** The first data category includes locomotion data from a study by ref. 6 from which we use data from quadrupeds of various species such as Virginia opossums (*Didelphis virginiana*), tufted capuchins (*Sapajus apella*), domestic dogs (*Canis lupus familiaris*), and the Australian water rat (*Hydromys chrysogaster*). Animals in this category locomote in a wide range of speeds, which correspond to different specific gaits. The animals were trained to sustain six to ten minutes of steady-state locomotion at any given speed, as required for metabolic measurements. A transition between walking and trotting gaits was observed by placing the animal onto an enclosed treadmill and incrementally increasing the speed of the moving belt every 15 s. The data has been extracted from the Supplementary Material of ref. 6.

**Extended gaits on flat terrain.** The second category comprises data from a study by Hoyt and Taylor[2] for walking and trotting gaits for horses, which we extracted from Fig. 2 [2] using the WebPlotDigitizer online tool. For given speed ranges, horses tend to locomote with a specific gait (i.e., walking at low speeds, trotting at medium speeds, and galloping at high speeds). However, in these experiments[2], horses were taught to extend their gaits for a wider range of speeds. For example, an extended trot is defined as a situation where a horse continues to trot at speeds above or below its normal trotting speed[41]. This is in contrast with the first category of data, where there is no gait extension and the animals locomote with their nominal gaits at all speeds.

## Interactions between the supraspinal drive and the central pattern generator
We use our hierarchical biology-inspired learning framework[39,47] for learning locomotion, as shown in Figs. 1 and. 7a. We have used the proposed scheme on flat terrain[47], as well as on a simple gap crossing scenario[39]. In this paper, we address the challenge of traversing consecutive gaps with small distances between them. Additionally, we explore the use of a height map around the robot in the observation space as an alternative to relying solely on explicit exteroceptive sensory features. The action space remains consistent with the simple gap-crossing scenario[39]. We extend and analyze the reward function, and in particular perform a systematic analysis on the importance of incorporating terms for viability (and velocity tracking terms), CoT, and peak contact forces. We formulate the supraspinal controller as an artificial neural network (ANN) which is trained with Deep Reinforcement Learning (DRL) to modulate the CPG intrinsic frequencies, amplitudes, and offsets of oscillation for each limb to coordinate and produce anticipatory behavior. The problem is represented as a Markov Decision Process (MDP), and we describe each of its components below. To train the policies, we use Proximal Policy Optimization (PPO)[58], a state-of-the-art on-policy algorithm for solving the MDP. Additional details can be found in the Supplementary Method 1.

**Action space.** We consider one RG layer for each limb based on Eqs. (1) and (2), where the RG output will be used in a PF layer to generate the spatio-temporal foot trajectories in Cartesian space (Eqs. (3) and (4)). Couplings between oscillators are known to exist in biological CPGs for coordinating gaits, but recent work has shown that they might be weaker than previously thought[22,24], and that sensory feedback and descending modulation might play an important role in inter-oscillator synchronization. We therefore investigate explicit coupling within the CPG dynamics, as well as implicit coupling through descending modulation from the supraspinal drive.

Flat Terrain: On flat terrain, our action space modulates the intrinsic amplitudes and frequencies of each oscillator which together forms the CPG, by continuously tuning $\mu_i$ and $\omega_i$ for each leg. We implement oscillator couplings representing both walking and trotting gaits through phase bias matrices ($\boldsymbol{\Phi}$) and coupling strengths ($w_{ij} = 1$). This coupling imposes a specific phase lag between oscillators and therefore constrains the policy to modulate parameters in order to locomote with these specific gaits.

Gap Crossing: For gap crossing scenarios, we do not consider explicit oscillator couplings ($w_{ij} = 0$), with the intuition that the terrain may prohibit certain gaits, and inter-limb coordination should thus be managed through the supraspinal drive. For gap crossing, in addition to modulating $\mu_i$ and $\omega_i$, we also consider modulating the oscillation set-points by learning $x_{off,i}$. Thus, our action space for gap-crossing can be summarized as $\mathbf{a} = [\boldsymbol{\mu}, \boldsymbol{\omega}, \mathbf{x}_{off}] \in \mathbb{R}^{12}$.

We divide the descending drive modulation into two categories: oscillatory components of the CPG dynamics $\mathbf{a}_{osc} = [\boldsymbol{\mu}, \boldsymbol{\omega}] \in \mathbb{R}^8$ and offset components $\mathbf{a}_{off} = [\mathbf{x}_{off}] \in \mathbb{R}^4$ which bypass the CPG dynamics, as in Eqs. (1)–(4).

**Observation space.** We consider two different observation space types based on (1) "blind" sensory information (enough for locomotion on flat terrain) and (2) also including exteroceptive anticipatory sensing for gap-crossing scenarios. We investigate visual exteroceptive information coming from several categories: (1) directly using visual depth information (i.e., LiDAR), (2) visually-extracted geometrical information (i.e., foot distance to a gap), and (3) instantaneous

feedback features (i.e., foot "penetration" into a gap). We investigate various combinations of these exteroceptive anticipatory features to understand the roles and importance of different sensory quantities to successfully cross variable gaps.

Blind (Flat Terrain) Observation: For locomotion on flat terrain, we consider sensory information that a "blind" agent could use to coordinate locomotion, in parallel with recent robotics works using DRL to learn legged locomotion[32,47]. This information includes vestibular sensory information (body orientation, body linear and angular velocity), proprioceptive sensory information (joint positions and velocities), foot contact booleans, the action chosen at the previous control cycle by the policy network, the internal CPG states, and the desired velocity command.

Exteroceptive-Visual Information: We consider two different methods for directly visually querying the surrounding environment. For the first method, we mount a LiDAR sensor at the front of the robot to return depth measurements along three channels (i.e., in PyBullet[59], Fig. 7a). We also consider querying the terrain heights within an area around the robot base, which can be constructed by geometrically transforming depth camera measurements from a camera mounted at the front of the robot (i.e., in Isaac Gym[60]).

Exteroceptive-Explicit Feedback Features: We assume that the visual system and brain can extract important geometrical information such as foot distance to a gap, and we call such information exteroceptive explicit feedback features. We are interested in investigating which explicit exteroceptive feedback features are most useful for the emergence of anticipatory locomotion skills. To reverse engineer this process, we divide the explicit sensory features into two categories: predictive and instantaneous feedback features. Predictive features consist of foot distance and/or base distance to the beginning and end of a gap. Instantaneous feedback features consist of boolean indicators of stepping into a gap (foot contact/penetration into the gap).

**Reward function.** We consider two locomotion scenarios: (1) steady-state locomotion on flat terrain with oscillator phase coupling for walking and trotting gaits, and (2) gap-crossing scenarios without oscillator coupling. To study the potential determinants of gait transitions, we propose a reward function that encompasses the following components in its general form (see Supplementary Method 2 for details):

- Viability: We promote viability by rewarding forward progress without falling. Locomotion speed during forward progress can be controlled by limiting (or penalizing) the reward for speeds above a maximum velocity threshold, or including a velocity tracking component. For the gap crossing scenario, we utilize the first approach, whereas for flat terrain locomotion, we employ the latter. Velocity tracking is implemented for flat terrain, as it necessitates running the policy at various velocities to measure the CoT. Accurately tracking a desired velocity promotes viability, as it ensures that the robot has not fallen while maintaining the desired task velocity. It is worth noting that the velocity tracking term is added to the forward progress term for flat terrain. For detailed information, please refer to the Supplementary Method 2.
- Cost of Transport: We penalize power in order to find energy efficient gaits. It is worth noting that, since the average velocity achieved by the robot remains consistent across all simulations, the CoT and power are roughly proportionally equivalent in these cases.
- Peak force: We penalize peak contact reaction forces in order to minimize body peak forces during locomotion.
- Base orientation penalty: We penalize body orientation deviations from a nominal horizontal position.

The importance of each component can be regularized by changing the weights. We systematically analyze the effects of the reward

function by considering four values of zero, low, medium, and high for the reward term weights for viability, CoT, and peak forces in gap-crossing scenarios. The low and medium weights are selected to be approximately 10% and 50% of the high value. Among all 64 possible combinations of zero-low-medium-high for viability, COT, and peak forces, the highest success rate is observed for the case with the highest viability and lowest CoT and peak contact force weights. We use high and low weight for the viability and energy efficiency terms for flat terrain locomotion. We do not incorporate the peak force component in the reward function for blind locomotion as our investigation on flat terrain focuses on walk-trot gaits. The reduction of musculoskeletal forces has been studied in the context of the trot-gallop gait transition in horses, particularly at high velocities[5]. However, for the walk and trot gaits at normal velocities, there is no significant presence of peak contact forces. The detailed weights of the reward function for each simulation result can be found in the Supplementary Table 4. Additionally, please refer to the Supplementary Method 1 and Supplementary Tables 1, 2 and 3 for details regarding the training method and settings.

### Training locomotion policies on flat terrain

On flat terrain, we train separate policies to learn specific gaits (walk and trot) by changing the oscillator phase coupling matrices ($\Phi$) and setting the coupling strength ($w_{ij} = 1$) in Eq. (1). Please refer to the Supplementary Method 6 for details regarding the phase coupling matrices. We assume that the supraspinal drive does not bypass the CPG dynamics to directly actuate joints in steady-state locomotion, so we do not modulate the offset components ($x_{off,i} = 0$) in Eq. (3). We use the blind (flat terrain) observation space for these experiments. We also define two separate training conditions regarding the two categories of animal data:

**Normal gait on flat terrain.** We train policies to test a wide range of speeds (i.e., $[v_{des,min}, v_{des,max}] = [0.3, 1.0] \frac{m}{s}$ for walking and $[v_{des,min}, v_{des,max}] = [0.9, 2.1] \frac{m}{s}$ for trotting) by resampling the desired velocity at the beginning of each environment reset. In order to ensure that the locomotion velocity varies in line with the stride frequency[6], the upper bound of the CPG frequency in the action space is expressed as a function of the desired velocity. Therefore, our action space limits are $\mu \in [0.5, 4]$, $\omega \in [0, f(v_{des})] (\frac{rad}{s})$, where $f(v_{des})$ is a linear function of the desired velocity:

$$f(v_{des}) = \frac{\omega_{max,2} - \omega_{max,1}}{v_{des,max} - v_{des,min}} * (v_{des} - v_{des,min}) + \omega_{max,1} \tag{5}$$

We use $\omega_{max,1} = 23$, $\omega_{max,2} = 60$ ($\frac{rad}{s}$) to train the walking policy, and $\omega_{max,1} = 30$, $\omega_{max,2} = 70$ ($\frac{rad}{s}$) to train the trotting policy. Therefore, during each episode, the upper bound of allowable frequencies is determined based on the desired velocity.

**Extended gait on flat terrain.** Biological studies have shown that horses continue to train and optimize their motor control through a lifelong learning process to locomote at certain limited speed ranges for each of their gaits[2]. To investigate locomotion principles, Hoyt and Taylor briefly trained horses to extend their gaits to speed ranges in which they would not normally use that gait[2] (i.e., trotting above/below their usual speed ranges). In order to increase their locomotion speed in an extended gait, the horses learned to increase their stride length. However, their stride frequencies remained approximately constant[41]. While horses learn to optimize their motor control policies over a long period of time, they were taught to extent their gaits during a short period, experiencing new locomotion parameters which they had not previously encountered in their lifetime.

In the context of robot locomotion, extending a gait can be understood as a scenario in which the robot is forced to locomote at a

speed for which it was not optimized for during the training process. To simulate experiments in this category, we trained two locomotion policies for certain limited speed ranges (i.e., $[0.4, 0.5]\frac{m}{s}$, for walking, and $[0.85, 0.95]\frac{m}{s}$ for trotting). To match the horse extended gait scenarios, we replicated the experiments with the robot by manually altering the stride length parameter ($L_{step}$ in Equation (3)) after training was complete, in order to increase or decrease the velocity. As a result, the policy observed parameter combinations in experiments with this new mapping which it had not encountered during the training process.

### Training locomotion policies for gap crossing scenarios
We investigate the robot's ability to learn to cross challenging terrains with multiple consecutive gaps. To allow the agent to coordinate behavior among different limbs through supraspinal drive, we do not consider explicit oscillator couplings ($w_{ij} = 0$ in Eq. (1)). We use the exteroceptive-visual information in addition to the blind terrain observation space. The action space has the following limits: $\mu \in [0.5, 4]$, $\omega \in [0, 40]$ $(\frac{rad}{s})$, $x_{off} \in [-7, 7]$ cm. The agent selects these parameters at 100 Hz, and they will therefore vary during each step according to sensory data.

In order to evaluate the different locomotion metrics (Fig. 5), we perform 14 policy rollouts (50,000 samples) on a test environment of locomoting at a desired velocity of 1 m/s over 4 randomized gaps. Each gap length is randomized between [14,20] cm during both training and test time, with 14 cm contact surface widths between gaps. An episode terminates early because of a fall, i.e., if the body height is less than 15 cm. We define the success rate as the number of gaps successfully crossed out of the total number of gaps. Additional training details can be found in Supplementary Method 1.

We train and test locomotion policies in both the PyBullet[59] and Isaac Gym[60] physics engine simulators. PyBullet[59] limits data collection to one robot per CPU core in our setup, while Isaac Gym[60] enables data collection from thousands of robots in parallel on a single GPU. Instead of LiDAR, Isaac Gym provides an interface to explicitly query the terrain heights in an area surrounding the robot body, which can be used to provide information about different types of discrete terrains. Fig.8a–d illustrates various snapshots of the Unitree A1 and Go1 robots locomoting on a variety of discrete terrains, including stepping stones, gap terrains with very small available contact surfaces, grid terrains, and mixtures of each these terrains, respectively. Despite these advantages, the precision of the gap position measurements in Isaac Gym is limited by the constraints of the mesh precision, which are imposed by the available GPU memory. In contrast, PyBullet enables us to easily obtain precise measurements of both the starting and ending positions of a gap. This capability is utilized to investigate the role of explicit exteroceptive sensory features, specifically the measurement of feet/base distance to a gap.

## Data availability
Data supporting the findings of this study are available within the paper and have been deposited in the Figshare database under accession code https://doi.org/10.6084/m9.figshare.23337158 (ref. 71). All other relevant data are available from authors upon reasonable request.

## Code availability
The CPG-RL framework implementation in Isaac Gym is available at https://github.com/MiladShafiee/DeepTransition (ref. 72).

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

## Acknowledgements

This research is supported by the Swiss National Science Foundation (SNSF) as part of project No.197237. M.Sh., and G.B. are supported by the Swiss National Science Foundation (SNSF) as part of project No.197237. We would like to thank Alessandro Crespi for assisting with hardware setup.

## Author contributions

M.Sh.: conceptualization, formal analysis, software, simulation, hardware experiment, investigation, writing original draft, visualization; G.B.: conceptualization, formal analysis, software, simulation, writing, review & editing; A.I.: conceptualization, formal analysis, review & editing, supervision, funding acquisition.

## Competing interests

The authors declare no competing interests.
