## [Peer Review File · Nature Communications]

REVIEWER COMMENTS

Reviewer #2 (Remarks to the Author):

This study proposes viability to be an important criterion for gait transitions, which is verified by both quadruped animal data on flat terrain and quadruped robots results on flat and gap terrains. The findings of the study suggest that viability could be a primary and universal objective of gait transitions. I think the work will be of significance to the related fields. The whole manuscript is well constructed and written. I have some minor questions:

1. How is viability quantified in this study? It was divided into high, medium, low as shown in Fig 4, but I can not directly find out how it is quantified in the manuscript.
2. Why were the low weights selected to be approximately 1% of the high values?
3. In Fig.3 b, it seems that the HL foot trajectory is higher than that of HR foot in XZ plane during pronk. Is there any explanations?

Reviewer #3 (Remarks to the Author):

This manuscript investigates the gait transition of a quadruped robot controlled by a deep reinforcement learning. The authors used viability as the learning criterion as well as general criteria, such as energy efficiency and stability. They performed computer simulations using two different environments (flat and gap terrains) and conclude that viability is a primary for the gait transition. The manuscript is well written and easy to understand. The concept of viability is interesting. However, I have some concerns as follows;

1. The gait transition in gap crossing is induced by a special environment with gaps, which is different from normal gait transitions to change the speed as explained in the beginning of the abstract.
2. In the special environment with gaps, it is difficult for robots to keep walking without falling, and thus it is reasonable that the viability is useful. However, it is unclear how viability works in normal environments.
3. The authors conclude that viability is useful even in flat environments. I guess that this is because the authors use a reinforcement learning where the robot falls down at the beginning of the training.

Although viability is useful for robots with a reinforcement learning, is it for robots without any reinforcement learning and animals that have already achieved gait?

4. The locomotion rhythm is generated by the oscillators and the desired robot motion is prepared. In addition, the robot uses many sensors to adapt to environments and disturbances. The robot can walk without the reinforcement learning. I am not sure to what the reinforcement learning contributed.

Specific comments are shown below.

L56: The definition of viability resembles that of basin of attraction.

Fig. 2A: The CV of animals looks constant regardless the speed and it increases in the gray regions. The CV of the robot looks different.

Fig. 2B: I am not sure why f , g , and h are shown here. There is no comparison with animal data.

L336-345: The relationship between viability and DCM offset is unclear.

Nature Communications 2023 - Response to reviewers

Manuscript NCOMMS-23-25203

Viability Leads to the Emergence of Gait Transitions in Learning Agile Quadrupedal Locomotion on Challenging Terrains

We would like to thank both reviewers for their constructive and insightful comments. Our first submission featured shortcomings in terms of analysis and discussion, particularly about the contribution of Deep Reinforcement Learning on flat terrain and quantification of the concept of viability. We have put our full efforts to improve the manuscript based on the excellent comments. We have shared our code and made it accessible in a private GitFront repository. The code is also available for download as a zip file through the corresponding PDF of the Description of Additional Supplementary Files. Upon publication of the paper, the code will be made publicly available. Moreover we have modified the title of the paper based on the editorial note. In the following we summarize the main changes and improvements. We hope that the reviewers will find our answers adequate and would welcome any additional feedback. This document combines both comments from, and answers to, the two reviewers.

The answers are formatted with color, while text changes are in the colored box formatted as follow:

"this text in the colored box has been re-written, ~~removed~~, or added."

Summary

Key Corrections:

- We have revised the title, abstract, section headings, and certain sections of the main text to adhere to the editorial policy and constraints.
- We have enhanced the precision and detail in our descriptions and explanations of the quantification of viability on flat terrain:
 - We have clarified the criteria used for quantifying viability on flat terrain in response to Comment 1.1 and Comment 2.2.
 - We have presented an analytical proof for the relationship between the divergent component of motion (DCM) offset and viability on flat terrain, based on the linear inverted pendulum model (LIPM) in response to Comments 2.1.2 and 2.1.3.
 - We have presented a new experiment to determine the maximum external push that the robot can tolerate without falling, thereby spotlighting viability on flat terrain.
- We have expanded the previous quantification of viability on gap terrain by training 64 neural network policies instead of the initial 27 policies, addressing Comment 1.2
- We have provided clarification regarding the role of the RL-trained neural network on flat terrain, in response to Comment 2.1.3 and 2.1.4.
- We have performed experiments to emphasize the role of the RL-trained neural network, in response to Comment 2.1.4.

- Due to word limitations, we have relocated most of the newly added materials to the Supplementary Information section.
 - We are pleased to inform you that we have provided the code and have made it open source.
-

1 Response to Reviewer 2

This study proposes viability to be an important criterion for gait transitions, which is verified by both quadruped animal data on flat terrain and quadruped robots results on flat and gap terrains. The findings of the study suggest that viability could be a primary and universal objective of gait transitions. I think the work will be of significance to the related fields. The whole manuscript is well constructed and written. I have some minor questions:

We would like to express our gratitude for your encouraging comments. It is our pleasure to make enhancements based on your insightful and constructive feedback.

1.1 Comment 1

1. How is viability quantified in this study? It was divided into high, medium, low as shown in Fig 4, but I can not directly find out how it is quantified in the manuscript.

Thank you very much for highlighting this point. In the proposed research Viability is **promoted** in the reward function during training of the locomotion policy, and then **quantified** and **evaluated** during test time by specific metrics. The value of low, medium and high is shown in the Supplementary Table 4. Here we highlight the corresponding reward function component for both flat terrain locomotion and gap crossing scenarios, and the specific metrics for quantifying them during test time:

- **Gap Crossing:** In the reward function, we promote forward progress in a gap crossing scenario which indicates that the robot is able to traverse the gaps without falling. The weights assigned to the reward function are detailed in the supplementary note of the paper. In accordance with the revised version based on the subsequent comment, we incorporate four different weight values: zero, low (10%), medium (50%), and high (100%). The assessment of viability in the gap-crossing scenario during test time involves quantifying the success rate of traversing variable environmental conditions. This can be evaluated, for instance, by determining the number of gaps the robot successfully crosses out of the total number of encountered gaps.
- **Flat Terrain:** In the reward function for learning locomotion on flat terrain, we have three terms that promote viability. *Forward progress* promotes viability of the system, as continuous high forward progress indicates that the robot has not fallen. *Velocity tracking* reward also promotes viability of the system, as tracking the steady state velocity indicates that the robot has not fallen. This term is already incorporated in the forward progress term; however, on flat terrain we would like to track different specific desired velocities to determine the relationship between the CoT and body velocity, as shown in Figure 2 of the paper. Large roll and pitch angles increase the likelihood of falling, thus *penalizing body orientation* helps to prevent the system from getting closer to the boundary of the viability kernel.

For quantifying the viability on flat terrain during test time, we use the following metrics: 1) **Divergent Component of Motion (DCM) offset**, 2) **Average Body Angular Velocity**, 3) **Coefficient of Variation (CV) of the stride duration** (as shown in Figure 2), and 4) **maximum external lateral force** (External force criteria added in the revised version). It has been shown that by increasing the DCM offset the system state approaches to the boundary of viability kernel, and therefore increases the risk of falling [1]. By increasing the locomotion speed, the increase of the DCM offset in the forward direction is inevitable since the movement will be more dynamic. However, the desired lateral DCM offset is zero, since in forward locomotion we do not have any desired lateral movement. Figure 2-f shows the reduction of the lateral DCM offset by performing walk-to-trot gait transition, therefore reducing the risk of falling. Granatosky et al. [2] propose that gait stability on flat terrain can also indirectly be evaluated by inter-stride

variability. Indeed, locomotion gaits with high inter-stride variability increase the risk of inter-limb interference, making tripping and/or falling more likely on flat terrain. This variability can be computed by the Coefficient of Variation (CV) of duration and is shown in Figure 2-a of the manuscript, where changing the gait from walk to trot reduces the CV of the stride duration. In the revised manuscript, we also investigate the maximum external push that the robot can tolerate without falling at each specific speed. The supplementary Figure 3 shows the maximum external pushes for each specific speed of locomotion on flat terrain, and we observe the curves are similar to the DCM offset and that the robustness against pushes improves by changing from a walk to trot gait. For the clarification, we added the following section to the methods section.

Supplementary Table 4: Reward function term weights for flat terrain and gap-crossing scenarios.

Reward function term weights				
Terms	Flat Terrain	Gap Terrain		
Viability	$\alpha_{1,flat} = 0.03$	$\alpha_{1,gap,low} = 0.8$	$\alpha_{1,gap,medium} = 4.0$	$\alpha_{1,gap,high} = 8.0$
Velocity Tracking	$\alpha_{2,flat} = 0.03$	×		
Peak Contact Force	×	$\alpha_{2,gap,low} = -0.001$	$\alpha_{2,gap,medium} = -0.005$	$\alpha_{2,gap,high} = -0.01$
Power	$\alpha_{3,flat} = -0.00001$	$\alpha_{3,gap,low} = -0.0001$	$\alpha_{3,gap,medium} = -0.0005$	$\alpha_{3,gap,high} = -0.001$
Base Orientation	$\alpha_{4,flat} = -0.02$	$\alpha_{4,gap} = -0.25$		

Method section, lines: 454-478:

Quantification of the Viability on Flat and Gap Terrain

In this paper, viability is promoted in the reward function during the training of the locomotion policy, and then quantified and evaluated during test time using specific metrics. Here, we summarize and recap specific metrics used for viability quantification during test time, along with the reward function components for both flat terrain locomotion and gap crossing scenarios.

Gap Crossing: In the reward function, we promote forward progress in a gap crossing scenario which indicates that the robot is able to traverse the gaps without falling. The weights assigned to the reward function are detailed in the supplementary note of the paper. We incorporate four different weight values: zero, low (10%), medium (50%), and high (100%). The assessment of viability in the gap-crossing scenario during test time involves quantifying the success rate of traversing variable environmental conditions. This can be evaluated, for instance, by determining the number of gaps the robot successfully crosses out of the total number of encountered gaps.

Flat Terrain: In the reward function for learning locomotion on flat terrain, we have three terms that promote viability. Forward progress promotes viability of the system, as continuous high forward progress indicates that the robot has not fallen. Velocity tracking reward also promotes viability of the system, as tracking the steady state velocity indicates that the robot has not fallen. This term is already incorporated in the forward progress term; however, on flat terrain we would like to track different specific desired velocities to determine the relationship between the CoT and body velocity, as shown in Figure 2. Large roll and pitch angles increase the likelihood of falling, thus penalizing body orientation helps to prevent the system from getting closer to the boundary of the viability kernel. For quantifying the viability on flat terrain during test time, we use the following metrics: 1) Divergent Component of Motion (DCM) offset, 2) Average Body Angular Velocity, 3) CV of the stride duration, and 4) maximum external force (as shown in Figure 2). It has been shown that by increasing the DCM offset the system state approaches to the boundary of viability kernel, and therefore increases the risk of falling [1]. Granatosky et al. [2] propose that gait stability on flat terrain can also indirectly be evaluated by inter-stride variability. Indeed, locomotion gaits with high inter-stride variability increase the risk of inter-limb interference, making tripping and/or falling more likely on flat terrain. We also investigate the maximum external push that the robot can tolerate without falling at each specific speed. The supplementary Figure 3 shows the maximum external pushes for each specific speed of locomotion on flat terrain. The DCM offset and maximum external push exhibit reasonable correlations for locomotion speed lower than 1.2 m s^{-1} (4.32 km h^{-1}).

1.2 Comment 2

2. Why were the low weights selected to be approximately 1% of the high values?

Thank you very much for your comment. Indeed, 1% of the high values can be considered an arbitrary choice, and possibly not a good one (not zero, yet very small). Therefore, we replaced it with low weight to be 10% of the high values, and also considered zero weight for each term. We trained new policies for the new combinations and updated Figure 4 of the paper accordingly. Now, instead of 37 cases, we have 64 cases with different neural network policies which have been trained based on all possible combinations of the reward function weights (0, 10%, 50%, 100%) for each of the three reward function terms (viability, COT, peak Forces). The qualitative behavior and conclusions are the same with these new results. We updated the corresponding part of “Emergence of Gait Transitions in Gap Crossing Scenarios” in the results section based on the new results.

Figure 4. Quantitative results from testing 64 policies with varying weights in reward function for crossing two series of 6 consecutive gaps. We consider three values of zero, low, medium, and high for the weights of the reward function terms for viability, CoT and contact peak forces. The zero, low and medium weights are selected to be approximately zero, 10% and 50% of the high values. We report average success rate, CoT, and contact peak forces for testing the policies for 4400 samples each (200 attempts of crossing 12 gaps). We show mean values and the standard deviations for success rate and Cost of Transport. Cases demonstrating similar behavior, like cases 1 to 16, have been grouped in the same column. To evaluate the peak forces, we first separately trained locomotion controllers on flat terrain to reach a velocity of 1.2 m s^{-1} , which is the average gap crossing velocity, and observed peak contact forces of 180N. For the gap crossing scenario, our reward function penalizes peak contact forces above this threshold at each control cycle, and the plots show the mean contact force in excess of 180N across all tests.

Results section, lines 202-238:

To analyze the impact of each component of the reward function, we trained 64 policies in

Isaac Gym, considering three different values (high, medium, low and zero) for the reward weights associated with viability, CoT, and contact peak forces. Figure 4 shows the success rate, CoT, and deviation of the contact force from its maximum value threshold across various cases. As exteroceptive sensory information, we include a height-map in front of the robot, beginning below its front hip (see Supplementary Material). Figure 4-a shows the weights of each case. The highest success rate is obtained with case 54, in which viability, CoT, and peak forces are assigned high, low, and low weights, respectively, as in the PyBullet results. It is noteworthy that this case exhibits the highest success rate of 99.25% across 2400 gap attempts, and interestingly the CoT remains approximately constant and CoT and peak forces increase after the gait transition from trotting on flat terrain to pronking over the discrete gap terrain. Case 23, with a high weight for viability and medium weights for CoT and peak forces, achieves the second highest success rate of 99.2%. Case 33, with a medium weight for viability and zero weights for CoT and peak forces, achieves the second highest success rate of 98.67%. However, the CoT and peak contact forces exhibit a deterioration after the transition. Among cases 10, 11, 20, and 22, 34, 37, 51 which all have a success rate above 98%, it is observed that in two cases the CoT decreases after the gait transition, while in case 37 it remains constant. The peak forces increase after the gait transition in all three of these cases. This suggests that the improvement of CoT and peak forces is not necessarily guaranteed after the gait transition. In cases 34 and 51, where the Cost of Transport (CoT) decreases after the transition, it is noteworthy that the weight assigned to CoT minimization in the reward function is zero. As a result, the trot gait, initially trained for walking before encountering the gap, proves to be less energy-efficient compared to other cases, such as case 37, where a higher weight is attributed to CoT minimization. The robot learns to walk with low reward function term values for all weights in case 1, however, it stops before the gap. In cases 2-9, by assigning a low weight to viability and various combinations of medium and high values for the contact peak forces and CoT, the robot remains stationary for the whole episode and does not exhibit any movement. For cases 1-16 which have zero viability weight, the robot remains stationary for the whole episode and does not exhibit any movement. Likewise, in cases 17 to 32 with a low weight for viability, all instances, except for 17 and 18, result in the robot remaining stationary. In cases 17 and 18, the robot learns to walk with a low reward weight for viability, zero weight for energy efficiency, and either low or zero weight for the contact force term. However, it comes to a halt before reaching the brink of the gaps, and does not cross them.

We observe an increase in peak forces, after the transition, in all cases which have a success rate higher than 50%. In case 36, with high weights for viability and peak contact forces, we observe that the robot learns to walk on flat terrain; however, it achieves a success rate of less than 50%. From case 56 onwards, where either high weights are assigned to contact forces or medium to high weights are assigned to CoT, a significant decrease in success rate is observed compared to cases 55, which have low and medium weights for CoT and contact peak forces respectively. In case 12, where medium and low weights are assigned for viability and CoT, and a high weight is assigned for contact forces, we observe a significant decrease in the success rate compared to cases 10 and 11, which have medium and low weights for contact peak forces. We observe a reduction in the overall CoT in cases 13 and 14, compared to cases 10 and 11. This reduction is attributed to the utilization of a medium weight for penalizing the CoT, albeit at the cost of a decrease in the success rate. In cases 12, 15, and 24, we observe a success rate of less than 50%, with the common characteristic of having a high weight for minimizing peak forces. Cases 16, 17, 18, 25, 26, and 27 fail to learn to move and instead remain stationary due to their high weight for penalizing the CoT.

1.3 Comment 3

3. In Fig.3 b, it seems that the HL foot trajectory is higher than that of HR foot in XZ plane during pronk. Is there any explanations?

Thank you very much for raising this point. Such an observation can be attributed to the fact that we do not constrain the locomotion policy to have symmetry in the system, and this is due to increased body roll movement after starting to cross the gaps. The following figure shows the body roll and pitch changes after crossing the gaps, and has been added to the supplementary figures of the paper. Moreover, the following text has been added to the main text of the paper:

Results section, lines 186-189:

HL foot trajectory is higher than that of HR foot in XZ plane. Such an observation can be attributed to the fact that we do not constrain the locomotion policy to have symmetry in the system, and this is due to increased body roll movement after starting to cross the gaps (Supplementary Figure 7).

Supplementary Figure 7: Body pitch and roll orientation are pertinent to the gap-crossing experiments depicted in Figure 3. The robot’s roll and pitch angles undergo abrupt changes upon reaching the gaps. In this scenario, the roll angle increases during gap crossing since there are no symmetry constraints imposed on the body. These roll angles elucidate why the Hind Left (HL) limb in Figure 3-b exhibits a greater height.

2 Response to Reviewer 3

This manuscript investigates the gait transition of a quadruped robot controlled by a deep reinforcement learning. The authors used viability as the learning criterion as well as general criteria, such as energy efficiency and stability. They performed computer simulations using two different environments (flat and gap terrains) and conclude that viability is a primary for the gait transition. The manuscript is well written and easy to understand. The concept of viability is interesting. However, I have some concerns as follows;

We appreciate your constructive comments and are pleased to enhance the paper based on your insightful feedback.

2.1 Major Comments

2.1.1 Comment 1

1. The gait transition in gap crossing is induced by a special environment with gaps, which is different from normal gait transitions to change the speed as explained in the beginning of the abstract.

Thank you for highlighting this crucial point. We concur that in the gap crossing scenario, the transition is induced by the gap environment. The initial sentence in the abstract was intended to emphasize the conventional aspects of gait transition present in the existing literature. However, we acknowledge that it might lead to confusion, and as a result, we have removed this sentence. We also would like to clarify that in the gap-crossing scenario, the gait transition is also accompanied with change of speed, as depicted in Figure 3-d of the paper. However, as noted in this comment, there is no intentional change of speed designed to trigger gait transition; instead, the inverse problem is observed, where gait transition leads to the change in speed. Therefore, we removed the following sentence from the abstract:

Abstract:

~~Quadruped animals seamlessly transition between gaits as they change locomotion speeds. Quadruped animals are capable of seamless transitions between different gaits. While energy efficiency appears to be one of the reason for changing gaits, other determinant factors likely play a role too, including terrain properties. the most widely accepted explanation for gait transitions is energy efficiency, there is no clear consensus on the potential determining factors, nor on the potential effects from terrain properties.~~

and we added the following sentence to the "*Emergence of Gait Transitions in Gap Crossing Scenarios*" section:

Results section, lines 194-195:

It is noteworthy that in this scenario, changes in speed do not induce the transition; rather, the gait transition results in a change of speed.

2.1.2 Comment 2

2. In the special environment with gaps, it is difficult for robots to keep walking without falling, and thus it is reasonable that the viability is useful. However, it is unclear how viability works in normal environments.

Thank you very much for your comment, we agree that this is a very important point necessitating further clarification. On the gap terrain, the gaps can be described as some potential disturbance that will lead to non-viable states. On the flat terrain, the potential disturbance can be described by external forces, or possibility of interfering of limbs, etc. There is the possibility that two different gaits avoid falls (i.e. are "viable") but have different level of viability. For example, the external pushes that robot with one gait is able to tolerate can be smaller than the other gait. This indicates that one of the gaits has a stricter viability condition and is near the boundary of viability kernel. The quantification of viability on flat terrain can be described by the Divergent Component of Motion (DCM) offset. As mentioned in the Methods section about Stride Duration Variability (Periodicity), viability on flat terrain can also indirectly be evaluated by inter-stride variability. Indeed, locomotion gaits with high inter-stride variability increase the risk of inter-limb interference, making tripping and/or falling more likely on flat terrain. So two gaits with both viable locomotion gait, one of them can have higher probability of inter-limb interference, and fall more easily in the event of a disturbance. **This point is further elaborated in the next comment, so please refer to comments 3 for the simulation results of various external pushes.**

2.1.3 Comment 3

3. The authors conclude that viability is useful even in flat environments. I guess that this is because the authors use a reinforcement learning where the robot falls down at the beginning of the training. Although viability is useful for robots with a reinforcement learning, is it for robots without any reinforcement learning and animals that have already achieved gait?

Thank you for bringing this point to our attention. In animal locomotion, Granatosky et al. [2] proposed that locomotion gaits with high inter-stride variability (high CV) increase the risk of inter-limb interference, making tripping and/or falling more likely on flat terrain. Consequently, in animal movements, the reduction of CV by changing the gait at the energetically optimal transition speed (EOTS) is closely related to viability. This reduction indicates that two gaits can have different viability statuses.

Throughout the flat terrain training process, reinforcement learning optimizes the neural network weights to discover actions that minimize nonviable states by promoting forward progress and therefore viability in the reward function. Following the training phase and during the subsequent testing of the learned gait, the significance of viability persists. This is attributed to the potential variance in viability conditions between the two gaits. For instance, we may have two viable gaits with distinct viability conditions. This can be clarified by the external push experiments, that show one gait can tolerate higher pushes without falling. For quantifying the viability of the robot on flat terrain during test time, we consider the Divergent Component of Motion (DCM) offset, shown in Figure 2-f of the manuscript. The

DCM concept is derived based on the Linear Inverted Pendulum Model (LIPM), demonstrating its ability to represent the dynamic locomotion of both quadrupeds and bipeds under certain assumptions [3, 4]. It has been shown that by increasing the DCM offset, the system state approaches the boundary of the viability kernel, and therefore increases the risk of falling [1]. By increasing the locomotion speed, the increase of the DCM offset in the forward direction is inevitable since increasing the CoM speed causes the DCM to also increase. However, the desired lateral DCM offset is zero, since during forward locomotion we do not have any desired lateral movement. Figure 2-f shows the reduction of the lateral DCM offset by changing from a walk to a trot gait. As stated in the "specific comments" section, the original paper lacked details on the relationship between the DCM offset and viability. To address this, we first present a viability analysis based on the DCM offset and the linear inverted pendulum model (LIPM). This analysis demonstrates that an increase in DCM offset beyond a certain threshold leads to the occurrence of non-viable states (and hence falls). However, this analysis is contingent on the assumptions of the Linear Inverted Pendulum Model (LIPM), including constant CoM height, zero change in angular momentum around CoM, rigid contact foot, and sufficient contact frictions. These assumptions may be violated in scenarios involving high-velocity locomotion [5] and in dynamical simulators that model contact as a spring/damper system. Additionally, the coefficient of friction may not be adequate to prevent slipping at high speeds. Therefore, we conducted external push experiments using a full model of quadrupedal simulation and determine the maximum external push that the robot can tolerate without falling at specific speeds.

We observe that the maximum external push maintains a consistent relationship with DCM offset during the gait switching events. Specifically, we observe a reduction in lateral DCM offset accompanied by an increase in the maximum push by walk-trot switches. However, for the trot gait, when the speed exceeds 1.3 m/s, we observe increases in both DCM offset and maximum lateral forces. This observation is inconsistent with the DCM offset viability analysis, and we attribute it to potential violations of the assumptions of the LIPM model at high speeds [5]. We have replaced Figure 2-g in the paper with external pushes, as the previous longitudinal DCM offset did not contribute significant information. Additionally, we have included Supplementary Figure 3 in the supplementary information. The subsequent sections have been added to the supplementary material of the paper and provide detailed insights into these analyses:

Supplementary information file, lines 82-146 :

Supplementary Method 3.1 Linear Inverted Pendulum

The LIPM has been widely utilized to describe the dynamics of the CoM for bipedal locomotion [3]. The LIPM assumes a constant rate of change of centroidal angular momentum and movement of the CoM height within a plane. Based on the assumptions, the equations of motion for the LIPM can be derived as follows:

$$\ddot{\mathbf{x}} = \omega^2(\mathbf{x} - \mathbf{cop}) \quad (1)$$

in which $\mathbf{x} = [x_{com}, y_{com}]^T$ is the horizontal position of the CoM, $\omega_0 = \sqrt{\frac{g}{\Delta z}}$ is the natural frequency of the LIPM, and $\mathbf{cop} = [cop_x, cop_y]^T$ is the horizontal position of the center of pressure (CoP).

Supplementary Method 3.2 Divergent Component of Motion (DCM) In this section, we provide an overview of the DCM concept's background. The dynamics of the CoM, as modeled by the LIPM, can be split into stable and unstable components. The unstable component is referred to as the DCM and is defined as follows:

$$\dot{\boldsymbol{\xi}} = \mathbf{x} + \frac{\dot{\mathbf{x}}}{\omega} \quad (2)$$

From (2), the CoM dynamics is given by:

$$\dot{\mathbf{x}} = \omega(\boldsymbol{\xi} - \mathbf{x}) \quad (3)$$

By differentiating (2) and substituting (1), the DCM dynamics is expressed as:

$$\dot{\xi} = \omega(\xi - \text{cop}) \quad (4)$$

Supplementary Figure 2 illustrates the relationship between the DCM dynamics, CoM, and the CoP. By re-arranging the DCM dynamics (4), the following ordinary differential equation (ODE) holds:

$$\dot{\xi} - \omega\xi = -\omega \text{cop}_0 \quad (5)$$

The solution to (5) can be written as:

$$\xi(t) = e^{\int \omega dt} \left[\int (-\text{cop}_0 \omega) e^{\int -\omega dt} dt + \mathbf{C} \right], \quad (6)$$

where $C \in \mathbb{R}^2$ is the vector of unknown coefficients that can be found by imposing the boundary conditions. Therefore, we can find these coefficients by solving the problem (6) either as an initial value problem, namely

$$\xi(0) = \xi_0 = \text{cop}_0 + \mathbf{C}_0, \quad (7)$$

or as a final value problem:

$$\xi(T) = \xi_T = \text{cop}_0 + \mathbf{C}_f e^{\omega T}. \quad (8)$$

Therefore, by solving the equation (4) as an initial value problem, we arrive at the following equation that represents the time evolution of the DCM:

$$\xi = (\xi_0 - \text{cop}_0) \exp(\omega t) + \text{cop}_0 \quad (9)$$

We can also solve the CoM dynamics (3) by treating it as an initial value problem:

$$\mathbf{x} = (\mathbf{x}_0 - \xi_0) \exp(-\omega t) + \xi_0 \quad (10)$$

As evident from the above equation, the CoM exhibits stable dynamics, with the exponential term being negative. However, the DCM exhibits unstable dynamics, characterized by a positive exponential term. This indicates that the difference between ξ_0 and cop_0 increases exponentially over time. The distance between the CoP and the DCM is referred to as the DCM offset, and minimizing this distance is crucial for maintaining viable states.

To find a DCM trajectory that satisfies both the initial and the final condition problems, the coefficient C_0 must equal C_f . Thus, by combining (7) and (8), we have:

$$\xi_0 - \text{cop}_0 = (\xi_T - \text{cop}_0) e^{-\omega T}. \quad (11)$$

Now by defining $\sigma = e^{\omega T}$ we obtain :

$$\xi_T + \text{cop}_0(-1 + \sigma) - \xi_0\sigma = 0. \quad (12)$$

Let cop_T represent the CoP position at the start of the next step, and $\gamma_T = \xi_T - \text{cop}_T$ denote the DCM offset for the next step (i.e, the end of this step) and $\gamma_0 = \xi_0 - \text{cop}_0$ denote the current DCM offset. Therefore, straightforward calculations lead to:

$$\gamma_T + \text{cop}_T - \gamma_0 \cdot \sigma = \text{cop}_0. \quad (13)$$

We will use this equation to determine the viability kernel for the LIPM.

Supplementary Method 3.3 Viability bound on the DCM offset

We now express the viability region of the LIPM [6, 1, 7, 8] in terms of the DCM offset. Computing the viability kernel is generally intractable, but fortunately it is possible to characterize these bounds for the LIPM as it was shown in [1]. It is noteworthy this viability analysis relies on the LIPM assumptions which assume a constant centroidal angular momentum, movement of the CoM height within a plane, and enough available coefficient of friction, which may not hold during high speed locomotion where the legs generate high angular momentum and the CoM height can have higher acceleration.

In equation (13), γ_T is the DCM offset for the next steps, cop_T is the CoP position for the next step and $L_{step} = \text{cop}_T - \text{cop}_0$ indicates the step length. Therefore, the maximum possible value for L_{step} is L_{max} , which is the maximum feasible step length of the robot. cop_0 indicates the current step position and current CoP. $\sigma = e^{\omega T}$ indicates the step duration, which is the fastest possible step of $\sigma_{min} = e^{\omega T_{min}}$, found based on actuation power of the robot.

By writing the (13) based on the step length we have:

$$L_{step} = -\gamma_T + \gamma_0 \cdot \sigma \quad (14)$$

We now describe the viability boundary using the DCM offset. We limit our analysis to the sagittal plane dynamics for forward walking as the analysis for backward walking is similar. The maximum DCM offset γ_{max} is linked to the maximum step length and minimum step duration by the following relationship:

$$\gamma_{max} = \frac{L_{max}}{\sigma_{min} - 1} \quad (15)$$

This maximum offset serves as a crucial threshold, distinguishing between viable and non-viable states: I) if the DCM offset is larger than γ_{max} , every potential combination of step timing and location will result in divergence and fall, II) if the DCM offset is smaller than (or equal to) γ_{max} , there is at least one possible combination of step timing and position that prevents the DCM from diverging (falling).

If $\gamma_0 > \gamma_{max}$ at the beginning of a step then we have:

$$\gamma_0 = \gamma_{max} + \epsilon \quad (16)$$

where $\epsilon > 0$. Using (14) and (16), for the DCM offset at the end of the step, we have:

$$\gamma_T = -L_{step} + \gamma_{max} \cdot \sigma + \epsilon \cdot \sigma \quad (17)$$

By substituting $L_{step} = L_{max}$ and $\sigma = \sigma_{min}$, we can determine the minimum achievable DCM offset $\gamma_{x,T}$:

$$\gamma_T = \gamma_{max} + \epsilon \cdot \sigma_{min} \quad (18)$$

Therefore, we observe that the minimum realizable DCM offset at the end of the step grows by $\epsilon \cdot \sigma_{min}$. Consequently, a series of steps will result in a diverging geometric series with a ratio of σ_{min} , implying that all possible choices of step location and timing will lead to divergence, ultimately resulting in a fall.

Supplementary information file, lines 146-181:

Robustness and Viability of Locomotion on Flat Terrain

In the preceding section, we noted that augmenting the DCM offset moves the system closer to the boundary of the viability kernel. Moreover, there exists a maximum DCM offset beyond which the system state becomes non-viable. Conversely, reducing the DCM offset moves the states of the system further inside the viability kernel, indicating an improvement in viability. Nevertheless, the DCM offset analysis in the preceding section relied on assumptions inherent to the linear inverted pendulum model (LIPM). The LIPM assumes zero change in angular momentum around the CoM, a constant CoM height, and enough available coefficient of friction. Although this model is suitable for low-speed locomotion, its applicability becomes more questionable for high-speed locomotion scenarios.

Viability on flat terrain for a quadruped robot can be quantified by the maximum external push the robot can tolerate without falling. For example, consider two walking gaits, both deemed viable. However, their robustness under external pushes may differ, with one being able to tolerate higher forces. This observation suggests that the latter gait exhibits better viability conditions. Our analysis of gait changes from walking to trotting on flat terrain revealed a reduction in lateral DCM offset. We hypothesize that the maximum external push a robot can withstand without falling will decrease after changing the gait at a specific speed. It is important to note that the DCM offset in the longitudinal direction will increase with speed for all gaits, as the CoM and DCM need to undergo faster and more dynamic forward locomotion. However, for forward locomotion, we desire a lateral DCM offset of zero since there is no intended lateral speed. In the following section, we will further explore the relationship between the DCM offset and external pushes. We

conducted simulations consisting of 350,000 samples, equivalent to an average of 50 tests of 7 seconds of locomotion each, applying external lateral pushes at various locomotion speeds for two policies trained for walking and trot gaits. In these simulations, a lateral external push was initiated at a random moment and lasts for 0.7 second. The falling state is identified when the robot reaches a height of less than 15 cm. The simulation is executed at various speeds, and the outcomes regarding the maximum applied force are illustrated in Supplementary Figure (3)-g. As evidenced by the transition from a walk to a trot gait at the energetically optimal speed, there is a noticeable increase in the maximum external force that the robot can tolerate without falling. This observation aligns with the changes in DCM offset, with a reduction in DCM offset after transitioning from a walk to a trot gait. This reduction signifies an improvement in viability after the gait transition. In the walk gait, as the speed increases, a consistent correlation is observed between the DCM offset and the maximum applied force. Conversely, in the trot gait, there is an overall reduction in the maximum force applied, aligning with the increasing DCM offset within the velocity range of 0.8 m s^{-1} to 1.2 m s^{-1} . However, beyond a velocity of 1.2 m s^{-1} , the maximum external forces show an increase that is not consistently in line with the escalating DCM offset during this speed interval. This discrepancy may be attributed to the assumptions inherent in the LIPM and DCM viability analysis, which rely on a zero centroidal angular momentum, constant body height, and available enough friction. These assumptions do not hold for high-speed velocities where limb rotation occurs rapidly.

Method section, lines 424-431:

To explore the viability condition of the full robot and validate the DCM offset analysis, we conducted external push experiments to assess the maximum push that the robots can withstand before falling. For each locomotion speed corresponding to Figure 2-f, we conducted multiple experiments, varying the force from 190 N to 400 N. Each force setting was tested for 350,000 samples across approximately 50 episodes, each lasting 7 seconds. The success rate corresponds to percentage of episodes where the robot completes the trial without falling. We then determined the highest force at which the robot achieved a 100 % success rate.

Results section, lines 143-161:

Since the concept of viability is difficult to quantify, we investigated three additional quantities which we think allow one to estimate whether one gait is more "viable" than another (Figure 2-B). The first quantity is the lateral Divergent Component of Motion (DCM) offset. Increasing the DCM generally corresponds to a more dynamic gait, which in turn increases the risk of falling. We observe that, as with the CV of the stride duration, switching gaits reduces the lateral DCM offset (panel (f)). The desired lateral DCM offset is zero, since the robot is walking forward in a straight line. The second quantity is the maximal lateral force that the robot can withstand before falling (panel (g)). Here again we see that the switch from walk to trot allows the robot to withstand higher forces. The third quantity is the average body angular velocity (panel (h)). Here we see that switching gaits decreases the average body angular velocity (panel (h)), improving stability. For all figures except (g), We report average values testing the policies for 35000 sample (7 tests of 5 s of locomotion) since the standard deviations are small (i.e. less than 10% of the mean), and the standard deviations are reported in the supplementary data. For the figure (g), the policies were tested for 350,000 samples, equivalent to an average of 50 tests of 7 seconds of locomotion each. In summary, our results show that on flat terrain, the transition from walking to trotting at a certain speed is not only useful to reduce the CoT, but also to increase the viability of the gait, making it more robust against lateral pushes, more periodic, with less angular velocity, and lower lateral DCM offset. Despite the enhanced viability achieved through the walk-trot transition, a comparison of figures "f" and "g" reveals that, in trot gait, when the speed exceeds 1.2 m s^{-1} (4.32 km h^{-1}), the maximum allowable push before falling decreases with an increasing DCM offset. This contradicts our viability analysis based on DCM offset. However, this outcome is somewhat anticipated, as the DCM offset is based on the linear inverted pendulum model, which relies on assumptions such as zero centroidal angular momentum, constant body height, and sufficient friction. These

assumptions do not hold for high-speed velocities, where limb rotation occurs rapidly.

Supplementary Figure 2: A: Inverted Pendulum and DCM, B: Viability Kernel [8]

2.1.4 Comment 4

4. The locomotion rhythm is generated by the oscillators and the desired robot motion is prepared. In addition, the robot uses many sensors to adapt to environments and disturbances. The robot can walk without the reinforcement learning. I am not sure to what the reinforcement learning contributed. Thank you for your comment. Reinforcement learning (RL) is responsible for training a neural network to continuously modulate the frequency and amplitude of the Central Pattern Generator (CPG) to adapt to sensory feedback. This adaptation is crucial for learning variable speed locomotion. Furthermore, adjusting the frequency and amplitude of the CPG during swing/stance phases, particularly in high-speed locomotion, is important. In particular, we conduct simulation studies where we fix the output of the RL-trained neural network as the input to the CPG. We use the trotting gait policy as shown in Figure (2)-a of the manuscript, setting the locomotion speed to 2.2 m s^{-1} . The neural network generates time-varying (based on sensory feedback) output representing the desired amplitude and frequency of the CPG. Subsequently, three random instances of this output are selected, and we fix the parameters of the CPG with these values so that they are no longer modulated. The simulation results indicate a failure to move forward in these cases, which highlight the contribution of the RL-trained neural network. To provide additional clarity on this aspect, we have included further text and simulations in the supplementary material.

Supplementary information file, lines 183-207:

The Role of RL in Quadrupedal Flat Terrain Locomotion

In this work, we employ Reinforcement Learning (RL) to train a neural network for modulating the frequency and amplitude of Central Pattern Generators (CPG). The primary objective of the Deep-RL approach is to optimize the actions of the deep neural network to maximize a reward function. In our specific context, the reward function on flat terrain aims to enhance viability while penalizing energy inefficiency and deviation from the desired velocity. In particular, the ability to generate viable locomotion behavior for different velocities using a single neural network underscores the pivotal role of Reinforcement Learning (RL). Furthermore, in scenarios involving high-speed locomotion, the continuous adaptation of CPG frequency and amplitude proves beneficial in coordinating phase changes during both stance and swing phases. To underscore the role of Reinforcement Learning (RL) in flat terrain locomotion within the proposed CPG-RL framework, simulations are conducted by fixing the output of the RL-trained neural network as the input to the CPG.

We use the trotting gait policy as shown in Figure (2)-a of the manuscript, setting the locomotion speed to 2.2 m s^{-1} . The neural network generates time-varying (based on sensory feedback) output representing the desired amplitude and frequency of the CPG. Subsequently, three random

Figure 2: Qualitative comparison data for robot and animal locomotion. **A:** The CoT and CV of stride duration of the animals and the robot are plotted against the locomotion speed of walk-trot gaits in: a) the quadruped robot, b) the domestic dog, c) the Australian water rat, d) the Virginia opossum and e) the tufted capuchin. This data relates to the first category of animal experiments with normal gaits from Granatosky et al [2]. The shadow vertical bars in (b)-(e) represent the range of preferred gait transition speeds for animals. In (a) and (f)-(g), the blue bar indicates the expected transition speed for the robot. The speed at which the CoT curve for a walking gait intersects with the CoT curve for a trotting gait represents the energetically optimal transition speed (EOTS). **B:** The lateral DCM offset, maximum allowable external push and average body angular velocity from the robot simulation in (a). **C:** The CoT is plotted against locomotion speed for the second category of animal data, i.e. horses with extended gaits from Hoyt and Taylor [9]. The shadow bar in (i) represents the speed at which horses prefer to locomote for that particular gait. In a lifelong learning process, this is the velocity for which the horse's motor control has been optimized for each gait [9]. In (j), the gray shadow bar represents the speed at which the policy is trained. The shaded area around the interpolated curves indicates the 95% confidence interval of interpolation. For the CV of stride duration for the animal data, Granatosky et al. [2] have calculated the mean of the stride duration for the specific speed interval. For external push in (g), we report the maximum lateral external force values for which the robot has a 100% success rate. Please note that the vertical axis of (g) is reversed for easier comparison with (f).

Supplementary Figure 3: In panels (f) and (g) of Figure 2 in the main text, the (f) shows the Lateral DCM Offset, while (g) illustrates the maximum lateral push that the robot can withstand before falling. Please note that the vertical axis of (g) is reversed for easier comparison with (f).

instances of this output are selected, and we fix the parameters of the CPG with these values so that they are no longer modulated. The simulation results indicate a failure to move forward in case 1, and falls in cases 2 and 3, which highlight the contribution of the RL-trained neural network.

In Supplementary Figure 4-a, the output of the neural network for the trotting policy is depicted, showcasing the desired amplitude and frequency of the CPG. The output of the neural network is normalized between -1 and 1, which is a common practice in deep reinforcement learning. Supplementary Figure 4-b presents snapshots of normal locomotion driven by the neural network and three distinct cases with fixed CPG inputs, illustrating locomotion failures with fixed CPG inputs. Furthermore, Supplementary Figure 5-a illustrates the constantly changing CPG phase and real amplitude, adapting based on the stance/swing phase of the limbs, where the shadow bar denotes the stance phase. Conversely, fixing the CPG frequency and amplitude results in a lack of adaptive behavior, as evidenced by the absence of such fluctuations.

The following text also added to discussion:

Discussion section, lines 294-297:

Overall, our study of flat terrain locomotion tends to show that both for animals and for our quadruped robot it is worth switching from walk to trot when increasing speed. The switch is beneficial both for increasing the energy efficiency of the gait (lowering the CoT) and for increasing the viability of the gait (e.g. increasing the robustness against lateral pushes).

Supplementary Figure 4: a. Output from the Reinforcement Learning-trained neural network, which corresponds to the neural network policy depicted in Figure (2)-a. The locomotion speed is set at $2.2 \text{ m}\cdot\text{s}^{-1}$. **b.** Cases 1, 2, and 3 are randomly selected, each representing a fixed desired CPG amplitude and frequency. This selection aims to explore the impact of the RL-trained neural network on locomotion. As illustrated in the snapshots of these cases, it is evident that the robot fails to achieve forward motion that highlights the importance of the RL on the flat terrain.

Supplementary Figure 5: **a.** The real CPG amplitudes and phases of various limbs, modulated by the Reinforcement Learning-trained neural network, contribute to the viability of locomotion at a speed of $2.2 \text{ m}\cdot\text{s}^{-1}$ (These plots correspond to the Supplementary Figure Figure 5-a). The shadow bar visually represents the stance phase of each limb, demonstrating the adaptive changes in phase based on the stance/swing mode of the limbs. **b.** CPG amplitude and phases of the different limbs of case 3 in Supplementary Figure 5. In this scenario, where the CPG parameters are fixed, the absence of adaptive behavior in the phase changes during the swing/stance phases results in nonviable locomotion, as demonstrated in Supplementary Figure 5-b.

2.2 Specific Comments

2.2.1 Comment 1

L56: The definition of viability resembles that of basin of attraction.

Thank you very much for raising this important point. We agree that viability and the basin of attraction share similarities in concept, however, having their differences.

An attractor's basin of attraction is the region of the phase space, over which iterations are defined, such that any point (any initial condition) in that region will asymptotically be iterated into the attractor. In the concept of bipedal locomotion, the basin of attraction represents the set of **initial conditions** or states from which the system, when perturbed slightly, will converge to a stable walking gait or pattern. This concept has mostly been investigated for the natural dynamics of the passive bipedal walker [10]. On the other hand, a viable state is guaranteed to have at least one sequence of **control actions** which, when applied from said state, will keep the system from failure [11]. Conversely, nonviable states are those where failure is no longer avoidable. The viability problem tries to answer the question of how to control dynamical systems subject to *viability constraints*. Therefore, viability has a close relation with the **available control inputs (action)**, but the basin of attraction focuses on the **initial conditions** of the system. Moreover, viability takes into consideration **environmental constraints**; for instance,

in the context of collision avoidance as a viability constraint, nonviable states are deemed Inevitable Collision States [12]. The following sentences have been added to the paper to clarify this point:

Methods section, lines 403-407:

"It is noteworthy that the basin of attraction and viability share similarities in concept; however, they also have their differences. In the topic of bipedal locomotion, the basin of attraction represents the set of initial conditions or states from which the system, when perturbed slightly, will converge to a stable walking gait or pattern [10]. Therefore, viability has a close relation with the available control inputs (action), but the basin of attraction focuses on the initial conditions of the system. Moreover, viability takes into consideration environmental constraints[12]."

2.2.2 Comment 2

Fig. 2A: The CV of animals looks constant regardless the speed and it increases in the gray regions. The CV of the robot looks different.

Thank you very much for this comment. The general reduction in coefficient of variation (CV), indicative of improved stability, is commonly observed when transitioning from a walk to a trot gait in both animals and robots. It is noteworthy that training two separate neural network policies for walk and trot gaits allows us to extend the gaits to velocities beyond the energetically optimal transition speed (EOTS). This is exemplified in the reduction of CV in the walk gait after the transition speed, a phenomenon not observed in animal data where, after the transition speed, there are no longer any walk gaits possible or available for the animals.

Furthermore, we acknowledge that detailed behaviors may differ between animals and robots due to distinct body elements and mechanics. However, the substantial reduction in the CV of stride duration by changing the gait around the energetically transition speed in robot demonstrates qualitative consistency with animal data (except possibly for the Australian rat for which there are very few data points). We added following text to clarify this:

Results section, lines 137-143:

It is noteworthy that training two separate neural network policies for walk and trot gaits allows us to extend the gaits to velocities beyond the EOTS. This is exemplified in the reduction of CV in the walk gait after the transition speed, a phenomenon not observed in animal data where, after the transition speed, there are no longer any walk gaits possible or available for the animals. Furthermore, we acknowledge that detailed behaviors may differ between animals and robots due to distinct body elements and mechanics. However, the substantial reduction in the CV of stride duration by changing the gait around the energetically transition speed in robot demonstrates qualitative consistency with animal data.

2.2.3 Comment 3

Fig. 2B: I am not sure why f, g, and h are shown here. There is no comparison with animal data.

Thank you for this comment. Regarding the animal data, we have information solely on the coefficient of variation of stride duration, which is linked to the concept of viability. In Granatosky et al.'s hypothesis, locomotion gaits with high inter-stride variability (higher CV of stride duration) are suggested to increase the risk of inter-limb interference, potentially leading to tripping or falling on flat terrain. Their measurement choice is based on the ease of calculation, as determining the center of mass (CoM) of the animal is challenging due to unclear mass distribution (and hence the DCM offset cannot be computed). Fortunately, our access to all states of the robot allows for the direct measurement of the DCM offset to analyze gait viability. Therefore, we have incorporated additional analyses and plots to illustrate the advantage of using robotics tools in investigating biological hypotheses.

2.2.4 Comment 4

L336-345: The relationship between viability and DCM offset is unclear.

Thank you for this important comment regarding the DCM offset and viability. We believe that addressing this point significantly enhances the understanding of the viability's role on flat terrain and its overall contribution. Given that the relationship between viability and DCM offset, as well as the role of viability on flat terrain, converge into a single problem, we have elucidated the entire discussion in the third major comment (3.1.3). In the response to comment 3.1.3, we conducted a viability analysis based on the DCM offset. It is demonstrated that there exists a DCM offset threshold, which if surpassed, makes a fall inevitable. This viability analysis is grounded in the linear inverted pendulum model (LIPM), which is a nice approximation of the locomotion dynamics. To validate this concept on the quadruped robot, an experiment involving external pushes on flat terrain has been conducted to elucidate the relationship between the DCM offset, viability, and robustness. The following sentence has been added to the text to refer the reader to the supplementary material:

Methods section, lines 424-426:

Please refer to the supplementary methods 3 and 4 for a detailed analysis of viability using the linear inverted pendulum model, as well as the interconnection between viability, DCM offset, and robustness during locomotion on flat terrain.

References

- [1] Majid Khadiv, Alexander Herzog, S Ali A Moosavian, and Ludovic Righetti. Walking control based on step timing adaptation. *IEEE Transactions on Robotics*, 36(3):629–643, 2020.
- [2] Michael C Granatosky, Caleb M Bryce, Jandy Hanna, Aidan Fitzsimons, Myra F Laird, Kelsey Stilson, Christine E Wall, and Callum F Ross. Inter-stride variability triggers gait transitions in mammals and birds. *Proceedings of the Royal Society B*, 285(1893):20181766, 2018.
- [3] Shuuji Kajita, Fumio Kanehiro, Kenji Kaneko, Kiyoshi Fujiwara, Kensuke Harada, Kazuhito Yokoi, and Hirohisa Hirukawa. Biped walking pattern generation by using preview control of zero-moment point. In *International Conference on Robotics and Automation*, volume 2, pages 1620–1626. IEEE, 2003.
- [4] Timothy M Griffin, Russell P Main, and Claire T Farley. Biomechanics of quadrupedal walking: how do four-legged animals achieve inverted pendulum-like movements? *Journal of Experimental Biology*, 207(20):3545–3558, 2004.
- [5] Shuuji Kajita, Hirohisa Hirukawa, Kensuke Harada, and Kazuhito Yokoi. *Introduction to humanoid robotics*, volume 101. Springer, 2014.
- [6] Milad Shafiee, Giulio Romualdi, Stefano Daffarra, Francisco Javier Andrade Chavez, and Daniele Pucci. Online dcm trajectory generation for push recovery of torque-controlled humanoid robots. In *2019 IEEE-RAS 19th International Conference on Humanoid Robots (Humanoids)*, pages 671–678. IEEE, 2019.
- [7] Pierre-Brice Wieber. On the stability of walking systems. In *Proceedings of the international workshop on humanoid and human friendly robotics*, 2002.
- [8] Pierre-Brice Wieber. Viability and predictive control for safe locomotion. In *2008 IEEE/RSJ International Conference on Intelligent Robots and Systems*, pages 1103–1108. IEEE, 2008.
- [9] Donald F Hoyt and C Richard Taylor. Gait and the energetics of locomotion in horses. *Nature*, 292(5820):239–240, 1981.
- [10] AL Schwab and M Wisse. Basin of attraction of the simplest walking model. In *International Design Engineering Technical Conferences and Computers and Information in Engineering Conference*, volume 80272, pages 531–539. American Society of Mechanical Engineers, 2001.
- [11] Jean-Pierre Aubin, Alexandre M Bayen, and Patrick Saint-Pierre. *Viability theory: new directions*. Springer Science & Business Media, 2011.
- [12] Mohamed Amine Bouguerra, Thierry Fraichard, and Mohamed Fezari. Viability-based guaranteed safe robot navigation. *Journal of Intelligent & Robotic Systems*, 95:459–471, 2019.

REVIEWERS' COMMENTS

Reviewer #2 (Remarks to the Author):

Thank you for your diligent efforts in addressing the suggested revisions to your manuscript. I have carefully reviewed your detailed responses and the corresponding modifications made in the revised manuscript. I am pleased to acknowledge that your thorough consideration of the feedback has significantly enhanced the overall quality of the paper.

Based on the comprehensive revisions made, I am of the opinion that the manuscript is now suitable for acceptance and publication in Nature Communications.

Reviewer #3 (Remarks to the Author):

The authors have addressed my concerns. I have no further comments.